**A new high-resolution pollen sequence at Lake Van, Turkey: Insights into penultimate interglacial-**
**glacial climate change on vegetation history**
Pickarski, N.[1], Litt, T.[1]
[1] *University of Bonn, Steinmann Institute for Geology, Mineralogy, and Paleontology, Bonn, Germany*
*Correspondence to:* Nadine Pickarski (pickarski@uni-bonn.de)
**Abstract**
A new detailed pollen and oxygen isotope record of the penultimate interglacial-glacial cycle,
corresponding to the Marine Isotope Stage (MIS) 7-6 has been generated from the 'Ahlat Ridge' (AR)
sediment core at Lake Van, Turkey. The presented Lake Van pollen record (c. 250.2-128.8 ka) displays
the highest temporal resolution in this region with a mean sampling interval of ~540 years.
Integration of all available proxies shows three temperate intervals of high effective soil moisture
availability, evidenced by the predominance of steppe-forested landscapes (oak steppe-forest) similar to
the present interglacial vegetation in this sensitive semi-arid region between the Black Sea, Caspian Sea,
and Mediterranean Sea.
The wettest/warmest stage as indicated by highest temperate tree percentages can be broadly correlated
with MIS 7c, while the amplitude of tree population maximum during the oldest penultimate interglacial
(MIS 7e) appears to be reduced due to warm but drier climatic conditions. The detailed comparison
between the penultimate interglacial complex (MIS 7) to the last interglacial (Eemian, MIS 5e) and the
current interglacial (Holocene, MIS 1) provides a vivid illustration of possible differences of successive
climatic cycles. Intervening periods of treeless vegetation can be correlated with MIS 7d and 7a, where
open landscape favour local erosion and detrital sedimentation. The predominance of steppe elements
(e.g., *Artemisia*, Chenopodiaceae) during MIS 7d indicates very dry/cold climatic conditions. In contrast,
the occurrence of higher temperate tree percentages (mainly deciduous *Quercus*) throughout MIS 7b
points to relatively humid and mild conditions, which is in agreement with other pollen sequences in
southern Europe.
Despite the general dominance of dry/cold desert-steppe vegetation during the penultimate glacial
(broadly equivalent to the MIS 6), this period can be divided into two parts: an early stage (c. 193-157 ka
BP) with higher oscillations in tree percentages, and a later stage (c. 157-131 ka BP) with lower tree
percentages and subdued oscillations. This subdivision of the penultimate glacial is also seen in other
pollen records from southern Europe (e.g., MD01-2444 and I-284; Margari et al., 2010; Roucoux et al.,
2011). The occurring vegetation pattern is analogous to the MIS 3 to MIS 2 division during the last glacial
in the same sediment sequence. Furthermore, we are able to identify the MIS 6e event (c. 179-159 ka BP)
as described in marine pollen records, which reveals clear climate variability due to rapid alternation in the
vegetation cover.
In comparison with long European pollen archives, speleothem isotope records from the Near East, and
global climate parameters (e.g., insolation, atmospheric $CO_2$ content), the new high-resolution Lake Van
record presents an improved insight into regional vegetation dynamics and climate variability in the
eastern Mediterranean region.

## 1.  Introduction

The long continental pollen record of Lake Van (Turkey) contributes significantly to the picture of long-
term interglacial-glacial terrestrial vegetation history and climate conditions in the Near East (Litt et al.,
2014). Based on millennial-scale time resolution (between c. 1-4 ka), the 600,000 year old pollen record
already shows a general pattern of alternating periods of forested and treeless landscapes that clearly
responds to the Milankovitch-driven global climatic changes (Berger, 1978; Martinson et al., 1987). In
that study, the Lake Van pollen record has demonstrated the potential ecological sensitivity for
paleoclimate investigations that bridge the southern European and Near East climate realms. Since then,
high-resolution multi-proxy investigations of the Lake Van sedimentary record have allowed the
systematic documentation of different climatic phases throughout the last interglacial-glacial cycle
(Pickarski et al., 2015a, 2015b).
To date, little attention has been focused on characterizing terrestrial sedimentary archives beyond 130 ka.
In particular, the detailed vegetation response to climatic and environmental changes in the Near East
during the penultimate interglacial-glacial cycle (Marine Isotope Stage (MIS) 7 to 6) has not been
thoroughly investigated.
In this context, we present new high-resolution pollen and oxygen isotope data from the 'Ahlat Ridge'
composite sequence over the penultimate interglacial-glacial cycle (between c. 242.5-131.2 ka). We have
added our recent results to the already existing low-resolution palynological and isotope data from Lake
Van published by Litt et al. (2014) and Kwiecien et al. (2014). This enables us to provide new detailed
documentation of multiple vegetation and environmental changes in eastern Anatolia by a centennial-to-
millennial-scale temporal resolution of ~180 to 780 years. Our record is placed in its regional context by
the comparison with several archives from the Mediterranean region, e.g., Lake Ohrid (between Former
Yugoslavian Republic of Macedonia and Albania; Sadori et al., 2016), Ioannina basin (NW Greece;
Frogley et al., 1999; Roucoux et al., 2008, 2011; Tzedakis et al., 2003a), Tenaghi Philippon (NE Greece;
Tzedakis et al., 2003b, 2006), and Yammoûneh basin (Lebanon; Gasse et al., 2011, 2015).
In our study, we address the following questions:

| 65 | (I) | What kind of regional vegetation occurred during the penultimate interglacial complex? Is the regional vegetation pattern of the oldest penultimate interglacial comparable to the last interglacial (Eemian) and current warm stage (Holocene)? |
| 66 | | |
| 67 | | |
| 68 | (II) | What processes characterized the climatic and environmental responses during the penultimate glacial? Is this vegetation history similar to the millennial-scale variability recorded during the last glacial in the same sequence? |
| 69 | | |
| 70 | | |
| 71 | (III) | Does the Lake Van vegetation history correlate with other existing long pollen records from southern Europe? What are the influencing factors of environmental change in the Near East? |
| 72 | | |

**Site description**
Lake Van is situated on the eastern Anatolia high plateau at 1648 m asl (meters above sea level; Fig. 1) in
Turkey. The deep terminal alkaline lake (~3574 km², max. depth >450 m) occupies the eastern
continuation of the Muş basin developed in the collision zone between the Arabian and Eurasian plates at
~13 Ma (Reilinger et al., 2006). Regional volcanism of Nemrut and Süphan volcanoes (at 2948 m asl and
4058 m asl, respectively; Fig. 1b), subaquatic hydrothermal exhalations and tectonic activities are still
active today, evident by the M 7.2 Van earthquake occurred on October 23, 2011 (Altiner et al., 2013).
The present-day climate at Lake Van is continental (summer-dry and winter-wet), with a mean annual
temperature of >9°C and mean annual precipitation between 400 and 1200 mm yr$^{-1}$ (Turkish State
Meteorological Service, 1975-2008; Table 1). In general, eastern Anatolia receives most of its moisture in
winter due to Cyprus low-pressure system within the eastern Mediterranean Sea (Giorgi and Lionello,
2008). At Lake Van, rainfall decreases sharply from south-west (c. 1232 mm a$^{-1}$ in Bitlis) to north-east (c.
421 mm a$^{-1}$ in Erciş; Table 1) due to orographic effects of NWW-SEE running Bitlis Massif parallel to the
southern shore of the lake (Fig. 1).
Due to the diverse topography at Lake Van, local variations in moisture availability and temperature are
quite pronounced, reflected in the modern vegetation distribution. At present, the vegetation cover around
Lake Van has been altered by agricultural and pastoral activities. According to Zohary (1973), the
southern mountain slopes are covered by the Kurdo-Zagrosian oak steppe-forest belt, containing *Quercus*
*brantii*, *Q. ithaburensis*, *Q. libani*, *Q. robur*, *Q. petraea*, *Juniperus excelsa*, and *Pistacia atlantica*. This
oak steppe-forest has also been described as 'mixed formation of cold-deciduous broad-leaved montane
woodland and xeromorphic dwarf-shrublands' by Frey and Kürschner (1989). In contrast, dwarf-shrub
steppes of the Irano-Turanian floral province is dominated by *Artemisietea fragrantis anatolica* steppe*,
different species of Chenopodiaceae, and grasses with some sub-Euxinian oak-forest remnants (Frey and
Kürschner, 1989; van Zeist and Bottema, 1991; Zohary, 1973).
**2. Material and methods**

## 2.1 Ahlat Ridge composite record

The sediment archive 'AR' (Ahlat Ridge; 38.667°N, 42.669°E at c. 357 m water depth; Fig. 1) was collected during the ICDP drilling campaign (International Continental Scientific Drilling Program, www.icdp-online.org) 'PALEOVAN' in summer 2010 (Litt and Anselmetti, 2014; Litt et al., 2012). The c. 219 mcblf (meter composite below lake floor) record contains a well-preserved partly laminated or banded sediment sequence, intercalated by several volcanic and event layers (e.g., turbidites; Stockhecke et al., 2014b). For further detailed description of the Lake Van lithology, we refer to Stockhecke et al. (2014b).

In this paper, we focus on a 60.1 m long sediment section from 117.19 to 57.10 mcblf representing the time span from c. 250.16-128.79 ka. In this section, we combine new pollen and isotope data with the already existing low-resolution pollen record published by Litt et al. (2014) and oxygen isotope data derived from bulk sediments ($\delta^{18}O_{bulk}$) analyzed by Kwiecien et al. (2014).

## 2.2 Chronology

The analytical approaches applied for the Lake Van chronology have previously been published in detail in Stockhecke et al. (2014a). All ages are given in thousands of years before present (ka BP), where 0 BP is defined as 1950 AD. Marine Isotope Stage (MIS) boundaries follow Lisiecki and Raymo (2004). Main results of the construction of the age-depth model are briefly summarized here.

For the investigated period, the age-depth model is based on independent proxy records, e.g., calcium and potassium element ratio (Ca/K) measured by high-resolution X-ray fluorescence (XRF; details in Kwiecien et al., 2014), total organic carbon (TOC; details in Stockhecke et al., 2014b), and pollen data (Litt et al., 2014). For the climatostratigraphic alignment of the presented Lake Van sequence, the proxy records were visually synchronized to the speleothem-based synthetic Greenland record (GL$_{T-syn}$ from 116 to 400 ka BP; Barker et al., 2011). The identifications of TOC-rich sediments containing high Ca/K intensities and increased AP (arboreal pollen) values at the onset of interstadials/interglacials were aligned to the interstadials/interglacial onsets of the synthetic Greenland record by using 'age control points'. Here, the correlation points of the Lake Van sedimentary record have been mainly defined by abiotic proxies (i.e., TOC) caused by a higher time resolution of this data set in comparison to the pollen samples available during that time. Even if we present a high-resolution pollen record in this paper, leads and lags between different biotic and abiotic proxies related to climate events have to be taken into account.

Furthermore, the age-depth model of the presented section (117.2-57.1 mcblf; 250.2-128.8 ka) was improved by adding two paleomagnetic time markers (relative paleointensity minima, RPI), analyzed by Vigliotti et al. (2014), at ~213-210 ka BP (Pringle Fall event; Thouveny et al., 2004) and at ~240-238 ka BP (Mamaku event; Thouveny et al., 2004). In addition, three reliable $^{40}Ar/^{39}Ar$ ages of single crystal

dated tephra layer at c. 161.9 ± 3.3 ka BP (V-114 at 71.48 mcblf), c. 178.0 ± 4.4 ka BP (V-137 at 82.29
mcblf), and c. 182 ka BP (V-144 at 87.62 mcblf; Stockhecke et al., 2014b) are used to refine the age-depth
model.

## 2.3  Palynological analysis

For the new high-resolution pollen analysis, 193 sub-samples were taken at 20 cm intervals. The temporal
resolution between each pollen sample, derived from the present age-depth model, ranges from ~180 to
780 years (mean temporal resolution c. 540 years).
Sub-samples with a volume of 4 cm³ were prepared using the standard palynological procedures by Faegri
and Iversen (1989), improved at the University of Bonn. This preparation includes treatment with 10% hot
hydrochloric acid (HCl; 10 min), 10% hot potassium hydroxide (KOH; 25 min), 39% hydrofluoric acid
(HF; 2 days), glacial acetic acid ($C_2H_4O_2$), hot acetolysis with 1 part concentrated sulfuric acid ($H_2SO_4$)
and 9 parts concentrated acetic anhydrite ($C_4H_6O_3$; max. 3 min), and ultrasonic sieving to concentrate the
palynomorphs. In order to calculate the pollen and micro-charcoal (>20 μm) concentrations (grains $cm^{-3}$
and particles $cm^{-3}$, respectively), tablets of *Lycopodium clavatum* spore (Batch no. 483216, Batch no.
177745) were added to each sample (Stockmarr, 1971). In all spectra, the average of ~540 pollen grains
was counted in each sample using a Zeiss Axio Lab.A1 light microscope. Terrestrial pollen taxa were
identified to the lowest possible taxonomic group, using the recent pollen reference collections of the
Steinmann Institute, Department of Paleobotany as well as Beug (2004), Moore et al. (1991), Punt (1976),
and Reille (1999, 1998, 1995). Furthermore, we followed the taxonomic nomenclature according to
Berglund and Ralska-Jasiewiczowa (1986).
Pollen results are given as a percentage and concentration diagram of selected taxa (Fig. 2). The diagram
includes the total arboreal pollen (AP; trees & shrubs) and non-arboreal pollen (NAP; herbs) ratio (100%
terrestrial pollen sum). In order to evaluate lake surface conditions, dinoflagellate cysts and green algae
(e.g., *Pseudopediastrum boryanum, P. kawraiskyi, Pediastrum simplex, Monactinus simplex*) were
counted on the residues from preparation for palynological analyses. Percent calculation, cluster analysis
(CONISS, sum of square roots) to define pollen assemblage zones (PAZ), and construction of the pollen
diagram were carried out by using TILIA software (version 1.7.16; ©1991–2011 Eric C. Grimm).
The complete palynological dataset is available on the PANGAEA database (www.pangaea.de;
https://doi.org/10.1594/PANGAEA.871228).

## 2.4  Oxygen isotope analysis

Stable oxygen isotope measurements ($\delta^{18}O_{bulk}$) were made on bulk sediment samples with an authigenic
carbonate content of ~30% ($CaCO_3$). Similar to the pollen analysis, 193 sub-samples were taken for the
new high-resolution isotope record at 20 cm interval within the penultimate interglacial-glacial cycle.
Before measurements were made, the samples were dried at c. 40°C for a least 48 hours and homogenized
by a mortar. The isotope analyses were carried out at the Leibnitz-Laboratory, University of Kiel, using a
Finnigan GasBenchII with carbonate option coupled to a DELTAplusXL IRMS.
All isotope values are reported in per mil (‰), relative to the Vienna Pee Dee Belemnite (VPDB)
standard. The standard deviation of the analyses of replicate samples is 0.02‰ for $\delta^{18}O_{bulk}$.
**3.    New data from the Lake Van sequence**
**3.1. The high-resolution pollen record**
The new palynological results from the penultimate interglacial-glacial cycle are illustrated in a simplified
pollen diagram (Fig. 2). Main characteristics of each pollen zone and the interpretation of their inferred
dominant vegetation types are summarized in Table 2.
The low-resolution pollen sequence, shown in Litt et al. (2014), has already been divided into six pollen
assemblage superzones (PAS IIIc, IV, Va, Vb, Vc, VI). This study followed the criteria for the
classification of the pollen superzones as described in Tzedakis (1994 and references therein). Based on
the new detailed high-resolution pollen sequence compared to the record in Litt et al. (2014), the PAS IV,
Va and Vc can now be further subdivided into 13 pollen assemblage zones (PAZ).
The pollen diagram provides a broad view of alternation between regional open deciduous oak steppe-
forest and treeless desert-steppe vegetation. We were able to recognized three main phases (PAZ Va1,
Va3, and during Vc2 and Vc3), where total arboreal pollen percentages reach above 30%. These phases
are predominantly represented by deciduous *Quercus* (max. ~56%), *Pinus* (max. ~26%), *Betula* (max.
~8%), and *Juniperus* (max. ~7%). However, AP maxima do not exceed 60-70%, suggesting that 'closed'
forest conditions were never established in eastern Anatolia. Mediterranean sclerophylls, e.g., *Pistacia* cf.
*atlantica*, are only present sporadically and at very low percentages. During open non-forested periods, the
most significant herbaceous taxa are the steppe elements Chenopodiaceae (max. ~76%), *Artemisia* (max.
~56%), and further herbs, such as Poaceae (max. ~54%), Tubuliflorae (max. ~13%), and Liguliflorae
(max. ~10%).
Throughout the sequence, the total pollen concentration values vary between c. 1700 and 52,000 grains
cm$^{-3}$. During PAZ IV1-6, Va2, Vb, and VI, the pollen concentration is dominated mainly by steppic
herbaceous pollen species (between 5000 and 52,000 grains cm$^{-3}$), whereas PAZ IIIc 6, Va1, Va3, and
Vc2-3 consist of tree and shrubs taxa (all above c. 5000 grains cm$^{-3}$).
In total, six green algae taxa were identified in the Lake Van sediments. Fig. 2a presents only the most
important *Pseudopediastrum* species. The density of the thermophilic taxa *Pseudopediastrum boryanum*
reached maxima values (c. 5500 coenobia cm$^{-3}$) combined with high AP percentages especially during
PAZ Vc2. In contrast, the cold-tolerant species *Pseudopediastrum kawraiskyi* occurred during treeless
phases (PAZ IV4-2; max. values c. 2000 coenobia $cm^{-3}$).
Furthermore, we calculated dinoflagellate concentration (probably *Spiniferites bentorii*; cysts $cm^{-3}$) in
order to get additional information about environmental conditions of the lake water (Dale, 2001;
Shumilovskikh et al., 2012). The occurrence of *Spiniferites* spp. in lacustrine sediments suggests low
aquatic bio-productivity (low nutrient level) and hypersaline conditions (Zonneveld and Pospelova, 2015;
Zonneveld et al., 2013). In this study, the concentration of dinoflagellate cysts is high (500-2000 cysts $cm^{-3}$)
during non-forested periods, especially within PAZ IV1, IV3, IV5, Va2, and PAS Vb (Fig. 2a).
The microscopic charcoal concentrations range between 300 and ~3000 particles $cm^{-3}$ during non-forested
phases when terrestrial biomass was relatively low (PAZ IV1-5, Va2, Vb and Vc1; Fig. 2a). During
forested phases, the charcoal content reaches maxima values of c. 8000 particles $cm^{-3}$ (e.g., in PAZ Va3,
Vc4-2).

## 3.2. The oxygen isotopic composition of Lake Van sediments

The general pattern of Lake Van isotope composition of bulk sediments shows very high-frequency
oscillation (Fig. 3). The $\delta^{18}O_{bulk}$ ranges from c. 5.9‰ to -4.6‰. Positive values occur between 250 and
244 ka, 238-222 ka, at 215 ka; 213-203 ka, 192-190 ka, 189-182 ka, and mainly between 171-157 ka and
141-134 ka. Negative isotope composition ($\delta^{18}O_{bulk}$ below 0‰) can be observed at ~241 ka; 221-216 ka;
202-194 ka; at ~181 ka, 178-171 ka, and between 156 and 155 ka.
Previous studies at Lake Van (e.g., Kwiecien et al., 2014; Lemcke and Sturm, 1997; Litt et al., 2012,
2009; Wick et al., 2003) have shown that the stable isotope signature of lake carbonates reflects complex
interaction between both several regional climatic variables and local site-specific factors. Such climate
variables are the moisture source, in this case the eastern Mediterranean Sea surface water and the storm
trajectories coming from the Mediterranean Sea, as well as temperature changes. Furthermore, the lake
water itself is related to the seasonality of precipitation (both rain and snowfall; water inflow) and
evaporation processes in the catchment area. However, the Lake Van authigenic carbonate $\delta^{18}O_{bulk}$ values
are primarily controlled by water temperature and isotopic composition of the lake water (T+$\delta^{18}O_w$;
Kwiecien et al., 2014; Leng and Marshall, 2004; Roberts et al., 2008).
At the beginning of terrestrial temperate intervals (e.g., PAZ Vc4, the end of Vb, Va1, and IIIc6), the
$\delta^{18}O_{bulk}$ composition of the lake water becomes more depleted (Fig. 3c). According to Kwiecien et al.
(2014) and Roberts et al. (2008), negative isotope values at the beginning of temperate intervals document
not only enhanced precipitation during winter months but also the significant contribution of depleted
snow melt/glacier meltwater during the summer months.

## 4. Discussion

## 4.1  Boundary definition and biostratigraphy

Based on long continental records in southern Europe (compiled by Tzedakis et al., 1997, 2001) and in the eastern Mediterranean area (Litt et al., 2014; Stockhecke et al., 2014a), it was shown that there is a broad correspondence between warm climatic intervals, respectively periods of low ice volume as defined by Marine Isotope Stages (MIS; Lisiecki and Raymo, 2004) and terrestrial temperate intervals (forested periods). In the continental, semi-arid Lake Van area it is difficult to use only the expansion of trees as criterion for the lower boundary of a warm stage. Therefore, the climatic boundaries at Lake Van were mainly defined by abiotic proxies (i.e., TOC) caused by a higher time resolution (Stockhecke et al., 2014a). However, we are aware that using different proxies do not necessarily occur at the same time (Sánchez Goñi et al., 1999; Shackleton et al., 2003). Even if we present a high-resolution pollen record in this paper, leads and lags between different biotic and abiotic proxies related to climate events have to be taken into account.

In addition, glacial/interglacial transitions (Termination) are near-synchronous global and abrupt climate changes. This scenario includes rising of Northern Hemisphere summer insolation, leading to ice-sheet melting and freshwater supply into the Atlantic Ocean (Denton et al., 2010). In this study, we follow the structure of Termination III at 250 ka, TIIIA at 223 ka, and TII at 136 ka after Barker et al. (2011) and Stockhecke et al. (2014a; Fig. 3, 5).

The climatostratigraphical term 'interglacial' and 'interstadial' were originally defined by Jessen and Milthers (1928) on the basis of paleobotanical criteria that are still generally accepted at present time. Here, an interglacial is understood as a temperate period with a climatic optimum at least as warm as the present-day interglacial (Holocene) climate in the same region. An interstadial is defined as a warm period that was either too short or too cold to reach the climate level of an interglacial in the same region. This definition is also valid for the Lake Van region as shown by Litt et al. (2014). In comparison, stadial stages correspond to cold/dry intervals marked by global and local ice re-advances (Lowe and Walker, 1984).

## 4.2  The penultimate interglacial complex (MIS 7)

According to Litt et al. (2014), the three-marked temperate arboreal pollen peaks (PAS Vc, Va3, and Va1) can be described as an interglacial complex. This general pattern of triplicate warm phases interrupted by two terrestrial cold periods (PAS Vb, PAZ Va2) is characteristic both in marine and ice-core records (MIS 7e, 7c, and 7a after Lisiecki and Raymo, 2004), as well as for continental pollen sequences in southern Europe correlated and synchronized by Tzedakis et al. (2001).

*Forested periods*

Within the penultimate interglacial complex, the three pronounced steppe-forested intervals PAS Vc
(113.7-109.1 mcblf, 242.5-227.4 ka), PAZ Va3 (104.2-101.3 mcblf, 216.3-207.6 ka) and PAZ Va1 (99.9-
97.0 mcblf, 203.1-193.4 ka) can be broadly correlated with the MIS 7e, 7c, and MIS 7a after Lisiecki and
Raymo (2004), indicating high moisture availability and/or warmer temperature (Fig. 2a, 3f).
The oldest terrestrial warm phase (242.5-227.4 ka, PAS Vc, MIS 7e) starts with the colonization of open
habitats by pioneer trees, such as *Betula*, followed by deciduous *Quercus* and sclerophyllous *Pistacia* cf.
*atlantica*. The occurrence of the frost-sensitive *Pistacia*, as a characteristic feature at the beginning of
interglacials in the eastern Mediterranean region, indicates relatively mild winters, but also firmly points
to the presence of summer aridity due to higher temperature and evaporation regime (Litt et al., 2014,
2009; Pickarski et al., 2015a; Wick et al., 2003). Similar to the Holocene, the early interglacial
spring/summer dryness might be responsible for the delay between the onset of climatic amelioration and
of the establishment of deciduous oak steppe-forest as the potential natural interglacial vegetation in
eastern Anatolia. Here, the length of the delay depending on local conditions keeping moisture availability
below the tolerance threshold for tree growth in the more ecologically stressed areas. Indeed, a reduction
of spring rainfall and extension of summer-dry conditions favoured the rapid development of a grass-
dominated landscape (mainly *Artemisia*, Poaceae; Fig. 2b). Furthermore, the fire activity rose at the
beginning of each warm phase when global temperature increased and the vegetation communities
changed from warm-productive grasslands to more steppe-forested environments. Increased fire frequency
is clearly visible by high charcoal concentration up to 3000 particles $cm^{-3}$ (Fig. 3e). After Termination III
at 243 ka, the vegetation change towards more steppe-forest environments correlates with depleted
(negative) $\delta^{18}O_{bulk}$ values, which occur at the beginning of the early temperate stage (c. 242-240 ka; Fig.
3c). As discussed earlier, depleted isotope values reflect intensified freshwater supply into the lake by
melting of Bitlis glaciers in summer months favouring high detrital input into the basin (low Ca/K ratio;
Fig. 3d) and/or enhanced precipitation during winter months (Kwiecien et al., 2014; Roberts et al., 2008).
The climate optimum of the first warm phase is characterized by significant expansion of temperate
summer-green taxa, mainly deciduous *Quercus* (above 20% between c. 240-237 ka), *Pistacia* cf. *atlantica*,
*Betula*, and sporadic occurrence of *Ulmus*. The vegetation composition documents a warm-temperate
environment with enhanced precipitation during the growing season, which can be supported by depleted
isotope values ($\delta^{18}O_{bulk}$ -2.17‰; Fig. 3c). Charcoal maxima (>3000 particles/cm³) correlates, coeval with
the delayed expansion of steppe-forest, with more fuel for burning. The gradual shift from depleted to
enriched isotope values ($\delta^{18}O_{bulk}$ 5.15‰) indicates a change towards climate conditions with high
evaporation rates and/or decreased moisture availability (Kwiecien et al., 2014; Roberts et al., 2008).
Here, positive $\delta^{18}O_{bulk}$ values at Lake Van are attributed to evaporative $^{18}O$-enrichment of the lake water
during the dry season. Furthermore, Kwiecien et al. (2014) described the relation between soil erosion
processes and vegetation cover in the catchment area. They defined interglacial conditions related to

increased precipitation indicated by higher amount of arboreal pollen and lower detrital input. Our new high-resolution pollen record validates their hypothesis with high authigenic carbonate concentration (high Ca/K ratio, low terrestrial input) along with the increased terrestrial vegetation density (high AP percentages above 50%) during the climate optimum (Fig. 3).

The ensuing ecological succession of the first warm stage is documented by a shift from deciduous oak steppe-forest towards the predominance of dry-tolerant and/or cold-adapted conifer taxa (e.g., *Pinus* and *Juniperus*; c. 237-231 ka). Especially, high percentages of *Pinus* suggest a cooling/drying trend, which occurred during low seasonal contrasts (low summer insolation and high winter insolation; Fig. 3). *Pinus* (probably *Pinus nigra*) as a main arboreal component of the 'Xero-Euxinian steppe-forest' recently occurs in more continental western and central Anatolia, and in the rain shadow of the coastal Pontic mountain range (van Zeist and Bottema, 1991; Zohary, 1973). Compared to the present distribution of *Pinus nigra* in Anatolia, the Lake Van region was probably more affected by an extended distribution area of pine during the penultimate interglacial as indicated by higher pollen percentages (Holocene below 5%; PAZ Vc2 up to 26%; PAZ Va3 up to 20%; Fig. 4). Holocene pine pollen was mainly transported over several kilometers via wind into the Lake Van basin. Independent of environmental conditions around the lake, the presence of thermophilic algae (i.e., *Pseudopediastrum boryanum*) displays warm and eutrophic conditions within the lake during the late temperate phase.

The presented regional vegetation composition can be described as an oak steppe-forest and marks one of the longest phases of the penultimate interglacial complex, lasting 15,000 years, with a climate optimum between 240 and 237 ka (Fig. 4c). However, this optimum does not appear of very high intensity as suggested by lower development of temperate plants compared to the following warm phase.

The second terrestrial temperate interval (PAS Vb-PAZ Va3; 106.5 -101.3 mcblf; c. 221-207 ka; MIS 7c) starts with a shift from cold/arid desert steppe vegetation (e.g., Chenopodiaceae) to less arid grassland vegetation (e.g., Poaceae, *Artemisia*; Fig. 2b). This was followed by an expansion of *Betula*, high abundance of deciduous *Quercus*, and continued with increased *Pinus* percentages. In this period, the occurrence of *Pistacia* cf. *atlantica* was not as pronounced as during the PAS Vc (MIS 7e), which can be explained by a lower winter insolation (cooler winters; Fig. 3b). Despite all this, the oxygen isotope signature displays similar depleted values ($\delta^{18}O_{bulk}$ up to -3.8‰; Fig. 3c) at the beginning of the middle warm phase, right after the Termination IIIA at 222 ka (Barker et al., 2011; Stockhecke et al., 2014a). In general, the second warm stage shows the highest amplitude of deciduous *Quercus* (peaked at 212.6 ka BP; Fig. 3f) of the entire sequence, which corresponds to the occurrence of the most floristically diverse and complete forest succession in southern European pollen diagrams at the same time (Follieri et al., 1988; Roucoux et al., 2008; Tzedakis et al., 2003b). In fact, deciduous *Quercus* percentages (c. 56%) reach the level of the last interglacial (MIS 5e) and the Holocene forested intervals, representing the most

humid and temperate period during the penultimate interglacial complex at Lake Van (Fig. 4; Litt et al.,
2014; Pickarski et al., 2015a).
Preliminary comparison with pollen records of Tenaghi Philippon (Tzedakis et al., 2003b) and Ioannina
basin (Roucoux et al., 2008) suggest that the extent and the diversity of vegetation development is clearly
controlled by insolation forcing and associated climate regimes (high summer temperature, high winter
precipitation). At Lake Van, the interglacial forest expansion is closely associated with the timing of the
Mid-June insolation peak (Tzedakis, 2005). In general, Mediterranean sclerophylls and other summer-
drought resistant taxa expanding during the period of max. summer insolation, while thermophilous taxa
are better suited to the less-seasonal climates of the later part of interglacial. Indeed, the highest expansion
of deciduous *Quercus* occurs, coeval to *Pinus*, during lowest seasonal contrasts (cooler summer and
warmer winters). The different amplitudes in the deciduous tree development might have resulted from
higher Mid-June insolation at the beginning of PAZ Va3 (MIS 7c) relative to PAZ Vc4 (MIS 7e, similar to
Holocene levels), despite lower atmospheric $CO_2$ content (c. 250 ppm, Fig. 5; Jouzel et al., 2007; Lang
and Wolff, 2011; Petit et al., 1999; Tzedakis, 2005), and thus, mirrored significant variability in regional
effective moisture content and/or temperature.
After a short-term climatic deterioration between 207 and 203 ka BP, the spread of *Pistacia* cf. *atlantica*,
*Betula*, and the predominance of deciduous *Quercus* characterize the youngest warm phase PAZ Va1
(99.9-97.0 mcblf, 203.1-193.4 ka, MIS 7a) within the penultimate interglacial complex. Similar to the
previous warm phases, the deciduous *Quercus* percentages (c. 38%) reach the level of the Holocene
forested interval (deciduous *Quercus* c. 40%; Fig. 4). A possible explanation for high thermophilous oak
percentages within MIS 7a is the persistence of relatively large tree populations through the cold period
equivalent to MIS 7b, which was also established in pollen records from Lac du Bouchet (Reille et al.,
2000) and at Ioannina basin (Roucoux et al., 2008).
All three forested stages of the penultimate interglacial complex are clearly recorded in other long
terrestrial pollen sequences from Lebanon and southern Europe: (I) the Yammoûneh record (Gasse et al.,
2015), (II) the Tenaghi Philippon sequence (Tzedakis et al., 2003b), (III) Ioannina basin (Roucoux et al.,
2008), and (IV) the Lake Ohrid sequence (Sadori et al., 2016). Fig. 5 shows that the Lake Van pollen
record generally agrees with the vegetation development of the Mediterranean region. However, we have
to take into consideration that most southern European sequences, e.g., the Ioannina basin, are situated
near to refugial areas, in which temperate trees persisted during cold stages (Bennett et al., 1991; Milner et
al., 2013; Roucoux et al., 2008; Tzedakis et al., 2002). In this places, where moisture availability was not
limiting, the woodland expansion occurred near the glacial/interglacial boundary (Tzedakis, 2007).
Despite this, high-resolution pollen records from the eastern Mediterranean region (e.g., Ioannina basin;
Roucoux et al., 2008) suggest that the MIS 7 winter temperature during all of these three warm intervals
seem to be lower than during the Holocene and the last interglacial as indicated by smaller populations of
sclerophyllous taxa. Reduced thermophilous components were also discussed for the Velay region (Reille
et al., 2000), where the warm phases Bouchet 2 and 3 equivalent to MIS 7c and 7a are described as
interstadials rather than interglacials. This observation of a cooler MIS in southern Europe contradicts to
the vegetation development at Lake Van, where all warm intervals reach the level of the last interglacial
and the Holocene. At Lake Van, there seems no reason to define the MIS 7c and MIS 7a as an interstadial,
separated from the MIS 7e interglacial.
*Non-forested periods*
The two periods between the three forested intervals, the first part of PAZ Vb (227-221 ka, 109.1-106.5
mcblf) and PAS Va2 (208-203 ka, 101.3-99.9 mcblf), are broadly equivalent to MIS 7d and MIS 7a
(Lisiecki and Raymo, 2004). At Lake Van, cold periods are generally characterized by: (I) extensive
steppe vegetation when tree growth was inhibited either by dry/cold or low atmospheric $CO_2$ conditions
(Litt et al., 2014; Pickarski et al., 2015b), (II) high dinoflagellate concentration (*Spiniferites bentorii,*
which tolerates high water salinity conditions and suggest low aquatic bio-productivity; Fig. 2a), and (III)
high regional mineral input derived from the basin slopes (low Ca/K ratio; Kwiecien et al., 2014; Fig. 3d).
Due to the strongest development of extensive semi-desert steppe plants (mainly Chenopodiaceae above
75%) and massive reduction of temperate tree (AP c. 5%; Fig. 2), the first cold phase suggests
considerable climate deterioration and increased aridity. Furthermore, this period is marked by large ice
volume and extremely low global temperatures, documented by low $CO_2$ concentration (~210 ppm; Fig. 5)
that are nearly as low as those of MIS 8 and 6 (McManus et al., 1999; Petit et al., 1999). Between 227 and
221 ka, the oxygen isotope record displays consistently $\delta^{18}O_{bulk}$ values above 0‰ that reflect dry climate
condition in the Lake Van catchment area (Fig. 3c). Such dry and/or cold period within the entire
penultimate interglacial complex can also be recognized in all pollen sequences from Lebanon and
southern Europe (Fig. 5; e.g., Gasse et al., 2015; Roucoux et al., 2008; Tzedakis et al., 2003b). An
exception is the Lake Ohrid record, which shows only a minor temperate tree decline (Sadori et al., 2016).
In contrast to conventional cold/dry periods at Lake Van, the second cold phase (PAS Va2) recognizes
only a slight and short-term steppe-forest contraction. Although the landscape was more open during the
youngest phase, moderate values of *Betula*, deciduous *Quercus* (up to 16%) and conifers (*Pinus*,
*Juniperus*) formed steppe vegetation with still patchy pioneer and temperate trees. The significantly larger
temperate AP percentages (c. 20%) during the PAZ Va2 relative to the PAZ Vb point to milder climate
conditions. In addition, the continuous heavier oxygen isotope signature ($\delta^{18}O_{bulk}$ between 1.0-2.4‰)
confirms the assumption of milder conditions with higher evaporation rates and more humid conditions.
Based on these results, the Lake Van pollen record mirrored the trend seen in various paleoclimatic
archives (Fig. 5). Indeed, several pollen sequences from the Mediterranean area and oxygen isotope
records suggest that the North Atlantic and southern European region (e.g., Ioannina basin; Roucoux et al.,
2008; Fig. 5d) did not experience severe climatic cooling during MIS 7b (e.g., Bar-Matthews et al., 2003;
Barker et al., 2011; McManus et al., 1999; Petit et al., 1999). In addition, the global ice volume remains
relatively low during the MIS 7b in comparison with other stadial intervals with similarly low insolation
values (e.g., Petit et al., 1999; Shackleton et al., 2000). Vostok ice-core sequence also records a relatively
high $CO_2$ content (c. 230-240 ppm) during MIS 7d supporting a slight decline of temperature compared
with MIS 7d ($CO_2$ content c. 207-215 ppm; Fig. 5; McManus et al., 1999; Petit et al., 1999).
*Comparison of past interglacials at Lake Van*
The direct comparison of the penultimate interglacial complex (MIS 7) with the last interglacial (Eemian,
MIS 5e; Pickarski et al., 2015a) and the current interglacial (Holocene, MIS 1; Litt et al., 2009) provides
the opportunity to assess how different successive climate cycles can be (Fig. 4).
In general, all interglacial climate optima were characterized by the development of an oak steppe-forest,
all of which reached the level of the last interglacial and the Holocene, especially the extent of temperate
tree taxa. Such dense vegetation cover reduced physical erosion of the surrounding soils in the lake basin.
Furthermore, the dominance of steppe-forested landscapes and productive steppe environment led to
enhanced fire activity in the catchment area. In addition to these aspects, the MIS 8/7e, MIS 7d/7c as well
as the MIS 6/5e boundary in the continental, semi-arid Lake Van region recognized a delayed expansion
of deciduous oak steppe-forest of c. 5000 to 2000 years, comparable to the pollen investigations in the
marine sediment cores west of Portugal by Sánchez Goñi et al. (2002, 1999). As already shown in high-
resolution pollen studies by Wick et al. (2003), Litt et al. (2009), and Pickarski et al. (2015a), a delay in
temperate oak steppe-forest refer to the Pleistocene/Holocene boundary as defined in the Greenland ice
core from NorthGRIP stratotype (for the Pleistocene/Holocene boundary; Walker et al., 2009) as well as
from the speleothem-based synthetic Greenland record ($GL_{T-syn}$; Barker et al., 2011; Stockhecke et al.,
2014) can be recognized. The length of the delay depending on slow migration of deciduous trees from
arboreal refugia (probably the Caucasus region) and/or by changes in seasonality of effective precipitation
rates (Arranz-Otaegui et al., 2017; Pickarski et al., 2015a). In particular oak species are strongly
dependent on spring precipitation (El-Moslimany, 1986). A reduction of spring rainfall and extension of
summer-dry conditions favoured the rapid development of a grass-dominated landscape (mainly
*Artemisia*, Poaceae; considered as competitors for *Quercus* seedlings) and *Pistacia* shrubs in the very
sparsely wooded slopes (Asouti and Kabukcu, 2014; Djamali et al., 2010). Furthermore, high intensity of
wildfires of late-summer grasslands, at the beginning of each warm period could be responsible for a
delayed re-advance of steppe-forest in eastern Anatolia (Arranz-Otaegui et al., 2017; Pickarski et al.,
2015a; Turner et al., 2010; Wick et al., 2003).
Despite the common vegetation succession from an early to late temperate stage, the three interglacial
periods (MIS 7 complex, MIS 5e, and MIS 1) differ in their vegetation composition. One important

difference of the last two interglacial vegetation assemblages is the absence of *Carpinus betulus* during MIS 7e, 7c, and 7a compared to a distinct *Carpinus* phase during MIS 5e (Pickarski et al., 2015a). In general, *Carpinus betulus* usually requires high amounts of annual rainfall (high atmospheric humidity), relatively high annual summer temperature, and is intolerance of late frost (Desprat et al., 2006; Huntley and Birks, 1983). In oak-hornbeam communities, *Carpinus betulus* is replaced as the soils are relatively dry and warm or too wet (Eaton et al., 2016). Compared to the common hornbeam, deciduous *Quercus* species are 'less' sensitive to summer droughts (even below 600 mm/a; Tzedakis, 2007), and therefore, a decrease in soil moisture availability would favor the development of deciduous oaks (Huntley and Birks, 1983). Especially, the deep penetrating roots of *Quercus petraea* allow them to withstand moderate droughts by accessing deeper water (Eaton et al., 2016). However, a variation in temperature is difficult to assess because deciduous oaks at Lake Van include many species (e.g., *Quercus brantii, Q. ithaburensis, Q. libani, Q. robur, Q. petraea*) with different ecological requirements (e.g., San-Miguel-Ayanz et al., 2016). Finally, the absence of *Carpinus betulus*, the overall smaller abundances of temperate trees (e.g., *Ulmus*), and the general low diversity within the temperate tree populations during the climate optimum of the first penultimate interglacial compared to the last interglacial indicates warm but drier climate conditions (similar to the Holocene). An exception is the second warm phase (MIS 7c), which reflects one of the largest oak steppe-forest development (e.g., highest amplitude of deciduous *Quercus*) of the entire Lake Van pollen sequence, and thus, represents the most humid and temperate period within the penultimate interglacial complex (see discussion above).

Another important difference is the duration of each interglacial period. According to Tzedakis (2005), the beginning and duration of terrestrial temperate intervals in the eastern Mediterranean region is closely linked to the amplitude of summer insolation maxima and less influenced by the timing of deglaciation. Based on this assumption, the terrestrial temperate interval of all penultimate interglacial stages (max. 15.1 ka) is ~4600 years shorter as the terrestrial temperate interval of the last interglacial at Lake Van (~ 19.7 ka, Pickarski et al., 2015a; Fig. 4).

**4.3 The penultimate glacial (MIS 6)**

The following penultimate glacial, PAS IV between 193.4-131.2 ka (58.1-96.8 mcblf), can be correlated with the MIS 6 (Lisiecki and Raymo, 2004; Fig. 2, 3). General lower summer insolation (Berger, 1978; Berger et al., 2007), increased global ice sheet extent (McManus et al., 1999), and decreasing atmospheric $CO_2$ content (below 230 ppm; Petit et al., 1999; Fig. 5) are responsible for enhanced aridity and cooling in eastern Anatolia. Such observed climate deterioration is suggested by the dominance of semi-desert plants (e.g., *Artemisia*, Chenopodiaceae) and by the decline in temperate trees (mainly deciduous *Quercus* <5%) similar to that of the last glacial at the same site. High erosional activity (low Ca/K ratio) and decreasing paleofire (Ø ~1400 particles $cm^{-3}$) result from low vegetation cover with low pollen productivity (Fig. 2,

3). As an additional local factor, the strong deficits in available plant water were possibly stored as
ice/glaciers in the Bitlis mountains during the coldest phases.
Between 193 and 157 ka BP, high-frequency vegetation (AP between ~1 and 18%) and environmental
oscillations (e.g., $\delta^{18}O_{bulk}$ values between -4 to 6‰) in the Lake Van proxies demonstrate a reproducible
pattern of centennial to millennial-scale alternation between interstadials and stadials, as recorded in the
Greenland ice core sequences for the last glacial (Fig. 3; e.g., NGRIP, 2004; Rasmussen et al., 2014). Such
changes indicate unstable environmental conditions with rapid alternation of slightly warmer/wetter
interstadials and cooler/drier stadials at Lake Van. In particular at 189 ka, the brief expansion of temperate
trees (deciduous *Quercus*, *Betula*) and grasses (Poaceae) combined with rapid variations in the fire
intensity (up to 6   000 particles cm$^{-3}$, Fig. 3e), decreasing terrestrial input of soil material (Fig. 3d), and
negative $\delta^{18}O_{bulk}$ values (-0.2‰) point to short-term humid conditions and/or low evaporation within
interstadials. Even if mean precipitation was low, the local available moisture was sufficient to sustain
arboreal vegetation when low temperature minimized evaporation. Nevertheless, the landscape around the
lake was still open due to still high percentages of dry-climate adapted herbs (e.g., Chenopodiaceae).
In contrast, the period after 157 ka BP shows a greater abundance of steppe elements with dwarf shrubs,
grasses and other herbs (e.g., Chenopodiaceae, *Artemisia*, *Ephedra distachya*-type) along with lower
temperate tree percentages (AP c. 1-8%). The remaining tree populations consist primarily of deciduous
*Quercus*, *Pinus*, with some scattered patches of *Betula* and *Juniperus*. The combination of minor AP
percentages, the predominance of steppe plants (Fig. 2b), and reduced fire activity reflect a strong
aridification and cold continental climate during the late penultimate glacial. In addition, a general low-
amplitude variation of $\delta^{18}O_{bulk}$ values (c. -2 to 2‰; Fig. 3b) and an overall high local erosion processes
(low Ca/K ratio; Fig. 3c) refer to a rather stable period with both widespread aridity (low winter and
summer precipitation) and low winter temperature across eastern Anatolia.
The Lake Van record generally agrees with high-frequency paleoenvironmental variations in the ice-core
archives, with high-resolution terrestrial European pollen records (e.g., Ioannina basin, Lake Ohrid; Fig.
5), and with the marine pollen sequences from the Iberian margin (Margari et al., 2010) in terms of
extensive aridity and cooling throughout the penultimate glacial. Our sequence also shares some features
with stable isotope speleothem records from western Israel (Peqi'in and Soreq Cave; Ayalon et al., 2002;
Bar-Matthews et al., 2003) concerning high $\delta^{18}O$ values that refer to dry climate conditions. Similar to the
Lake Van $\delta^{18}O_{bulk}$ values, the Soreq and Peqi'in record also show distinct climate variability, especially at
the beginning of the MIS 6 (Fig. 5). In addition, several high-resolution terrestrial records document a
further period of abrupt warming events between 155-150 ka BP. In particular, the Tenaghi Philippon
profile illustrates a prominent increase of up to 60% in arboreal pollen, which coincides with increased
rainfall at Yammoûneh (Gasse et al., 2015) and at Peqi'in Cave (Bar-Matthews et al., 2003). At Lake Van,
only a weakened short-term oscillation can be detected in the Ca/K ratio during that time.

### *Comparison of the last two glacial intervals at Lake Van*

The occurrence of high-frequency climate changes within the Lake Van sediments provides an opportunity to compare the vegetation history of the last two glacial periods. Fig. 6 illustrates that the first part of the penultimate glacial (c. 193-157 ka) resembles MIS 3, regarding millennial-scale AP oscillations and abruptness of the transitions in the pollen record. The series of interstadial-stadial intervals can be recognized in both glacial periods. This variability is mainly influenced by the impact of North Atlantic current oscillations and the extension of atmospheric pattern, in particular, northward shift of the polar front in eastern Anatolia (e.g., Cacho et al., 2000, 1999; Chapman and Shackleton, 1999; McManus et al., 1999; Rasmussen et al., 2014; Wolff et al., 2010).

The most distinct environmental variability occurred during MIS 6e (c. 179-159 ka), which can be further divided into six interstadials based on rapid changes in the marine core MD01-2444 off Portugal (Margari et al., 2010; Roucoux et al., 2011; Fig. 6). They document abrupt climate oscillations below orbital cycles similar to the Dansgaard-Oeschger (DO) events or Greenland Interstadials (GI) over the last glacial stage (e.g., Dansgaard et al., 1993; Rasmussen et al., 2014; Wolff et al., 2010). At Lake Van, the MIS 6e reveals a clear evidence of climate variability due to rapid alternation in abiotic and biotic proxies such as oxygen isotopes, Ca/K ratio, and pollen data similar to the largest DO 17 to 12 during MIS 3 (c. 60-44 ka BP; Pickarski et al., 2015b). Both intervals, MIS 6e and MIS 3, started at the point of summer insolation maxima. Here, the Northern Hemisphere insolation values reached interglacial level at the beginning of MIS 6e comparable with MIS 7e (Fig. 5). In contrast, the interstadial-stadial pattern during the late MIS 6 oscillated at lower amplitude, similar to rates of change in the Dansgaard-Oeschger (DO) events during MIS 4 and 2, reflecting a general global climatic cooling.

Within the MIS 6e, the subdued temperate tree pollen oscillations consist mainly of deciduous *Quercus* and *Pinus*, range between ~1 and 15%. In contrast, the identical AP composition oscillates between ~1 and 10% during the orbitally equivalent MIS 3 (c. 61-28 ka; Pickarski et al., 2015b). The different amplitude in arboreal pollen percentages in both glacial stages and a general dense temperate grass steppe during the MIS 6e suggest more available moisture (Fig. 6). Depleted isotope signature may result from summer meltwater discharge from local glaciers (e.g., Taurus mountains, Bitlis Massif) or by increased precipitation identified by climate modeling experiments over the eastern Mediterranean basin (e.g., Stockhecke et al., 2016). However, the presence of *Artemisia* and Poaceae makes it difficult to disentangle the effects of warming from changes in moisture availability in both glacials. Nevertheless, the abundance of *Pinus*, *Ephedra distachya*-type as well as the cold-tolerant algae *Pseudopediastrum kawraiskyi* indicates colder/wetter climate conditions during MIS 6e compared to MIS 3.

Evidence for relatively humid but cold climate conditions during MIS 6e agrees with several other paleoclimate studies from the Mediterranean area. For example, the occurrence of open forest vegetation

associated with wetter climate is indicated at, e.g., Tenaghi Philippon (Tzedakis et al., 2006, 2003b) and Ioannina (Roucoux et al., 2011). In addition, isotopic evidence of the stalagmites record from the Soreq Cave (Israel) shows enhanced rainfall (negative shift in the $\delta^{18}O$ values) in the eastern Mediterranean at ~177 ka and between 166-157 ka BP (Fig. 5; Ayalon et al., 2002; Bar-Matthews et al., 2003). Furthermore, a pluvial phase is also inferred from a prominent speleothem $\delta^{18}O$ excursion in the Argentarola Cave (Italy) between 180 and 170 ka BP based on U/Th dating (Bard et al., 2002). This phase coincides with maximum rainfall conditions during MIS 6.5 event, coeval with the deposition of the 'cold' sapropel layer S6 (c. ~176 ka BP) in the western and eastern Mediterranean basin (Ayalon et al., 2002; Bard et al., 2002). Finally, the progressive decline in effective moisture is a result of the combined effect of temperature, precipitation and insolation changes in the Lake Van region.

## 5. Conclusions

1. The new high-resolution Lake Van pollen record provides a unique sequence of the penultimate interglacial-glacial cycle in eastern Anatolia (broadly equivalent to the MIS 7 and MIS 6) that fills the gap in data coverage between the northern Levant and southern Europe. It reveals three steppe-forested intervals that can be correlated with MIS 7e, 7c, and 7a. Intervening periods of more open, herbaceous vegetation are correlated with MIS 7d and 7b.

2. During the penultimate interglacial complex, high local and regional effective soil moisture availability is evident by a well-developed temperate oak steppe-forest with pistachio and juniper, high charcoal accumulation, and reduced physical erosion during the climate optima.

3. In contrast to south-western Europe, all three terrestrial warm intervals of MIS 7 are characterized by clear interglacial conditions. The largest oak steppe-forest expansion in the Lake Van region within the penultimate interglacial complex occurred during the terrestrial equivalent of the MIS 7c instead of MIS 7e. This underlines the different environmental response to global climate change in the continental setting of the Near East compared to global ice volume and/or greenhouse gas.

4. The eastern Mediterranean Lake Van pollen sequence is in line with data from long-term climate records from southern Europe and the northern Levant, in terms of vegetation changes, orbitally-induced fluctuations, and atmospheric changes over the North Atlantic system. However, the diversity of tree taxa in the Lake Van pollen spectra seems to be rather low compared to southern European terrestrial interglacials and their forest development.

5. During the penultimate glacial, strong aridification and cold climate conditions are inferred from open desert-steppe vegetation that favors physical erosion and local terrigenous inputs. In particular, our record reveals high temperate oscillations between 193-157 ka BP, followed by a

period of lower tree variations and the predominance of desert-steppe from 157-131 ka BP that

highlighted Dansgaard-Oeschger-like events during the MIS 6.

**Data availability:** The complete pollen data set is available online on the PANGAEA database
(https://doi.org/10.1594/PANGAEA.871228).
**Acknowledgements**
Financial support was provided by the German Research Foundation (DFG; LI 582/20-1). We thank all
colleagues and scientific teams who have been involved in the Lake Van drilling, core opening and
sampling campaigns. We thank Dr. Nils Andersen and his working team at the Leibnitz-Laboratory for the
isotopic measurements. We acknowledge Vera Pospelova and Fabienne Marret-Davies for their help to
identify dinoflagellate cysts. We thank Karen Schmeling for preparing excellent pollen samples, Christoph
Steinhoff and Helen Böttcher for their support in the lab. Special thanks go to Ola Kwiecien and Georg
Heumann for their critical reading of the manuscript and for the inspiring discussions. Patricia Pawlyk, as
a native speaker, is thanked for proof reading the English. We are grateful to Miryam Bar-Matthews and
Avner Ayalon from the Geological Survey of Israel (Jerusalem) for the supply of the oxygen isotope data
of the Soreq and Peqi'in record. The authors are grateful to Nathalie Combourieu-Nebout for editing of
the manuscript. Donatella Magri, Gonzalo Jiménez-Moreno and two anonymous reviewers are
acknowledged for their constructive comments and useful recommendations, which improved the quality
of the manuscript.

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

Zohary, M., 1973. Geobotanical Foundations of the Middle East. Gustav Fischer Verlag, Swets &
Zeitlinger. Stuttgart, Amsterdam.


 **Figures**

**Fig. 1:** Map of the eastern Mediterranean region showing major tectonic structures in Turkey. (a) Location
of key Mediterranean and Near East pollen sites (stars) and speleothem records (triangle) mentioned in the
text. (b) Bathymetry of Lake Van including the Ahlat Ridge drill site (AR, star). The black triangle
indicates the positions of the active Nemrut and Süphan volcanoes. NAFZ: North Anatolian Fault Zone;
EAFZ: East Anatolian Fault Zone; BS: Bitlis Suture.
**Fig. 2:** Pollen diagram inferred from Lake Van sediments plotted against composite depth (mcblf) and age
(ka BP). (a) Selected arboreal pollen abundances are expressed as percentages and concentrations of the
pollen sum (black curves), which excludes bryophytes, pteridophytes, and aquatic taxa. Rare taxa are
summed and presented as 'Other AP'. Selected arboreal pollen concentration (grains per cm³; red bars) is
also given. Concentrations of green algae (*Pseudopediastrum boryanum*, *P. kawraiskyi*, coenobia per cm³;
black bars), dinoflagellates (cysts per cm³; black bars), and charcoal particles (>20 μm, particles per cm³;
black bars) are presented. (b) Selected pollen percentages diagram for non-arboreal taxa and key aquatic
herbs (grey curves). Percentages and concentrations are calculated as for arboreal pollen. Rare taxa are
summed as 'Other NAP'.
Pollen assemblage superzones (PAS) and zones (PAZ, grey dashed lines) are indicated on the right and
described in Table 2. Intervals characterized by oak steppe-forest (AP >30%) are marked in each diagram
(grey box). An exaggeration of the pollen curves (x10; white curves) is used to show low variations in
pollen percentages.
**Fig. 3:** Comparative study of Lake Van paleoenvironmental proxies during the penultimate interglacial-
glacial cycle. (a) LR04 isotopic record (in ‰ VPDB) with Marine Isotope Stage (MIS) boundaries (grey
bars) following Lisiecki and Raymo (2004); (b) Insolation values (40°N, $Wm^{-2}$) after Berger (1978) and
Berger et al. (2007); (c) Lake Van oxygen isotope record $\delta^{18}O_{bulk}$ (‰ VPDB; new analyzed isotope data
including the already published isotope record by Kwiecien et al., 2014); (d) Calcium/potassium ratio
(Ca/K) after Kwiecien et al. (2014); (e) Fire intensity at Lake Van (>20 μm, charcoal concentration in
particles $cm^{-3}$); (f) Selected tree percentages (total arboreal pollen (AP), deciduous *Quercus*, and *Pinus*)
including the pollen data from Litt et al. (2014). PAZ – Pollen assemblage zone. Termination III at 250 ka,
TIIIA at 223 ka and TII at 136 ka are indicated after Barker et al. (2011) and Stockhecke et al. (2014a).
**Fig. 4:** Comparison of (a) current interglacial (MIS 1; Litt et al., 2009) with (b) last interglacial (MIS 5e;
Pickarski et al., 2015a), and (c) penultimate interglacial complex (MIS 7; this study) at Lake Van. Shown
is the insolation values (40°N, $Wm^{-2}$) after Berger (1978) and Berger et al. (2007), the Lake Van arboreal
pollen (AP) concentration (grains $cm^{-3}$, brown line), and the Lake Van paleovegetation (AP, deciduous
*Quercus*, and *Pinus* in %). The grey boxes mark each steppe-forest intervals. Marine Isotope Stage (MIS;
Lisiecki and Raymo, 2004) and the length of each interglacial (MIS 5e, 7a, 7c, and 7e, black arrows) are
indicated.
**Fig. 5:** Comparison of Lake Van pollen archive with terrestrial, marine and ice core paleoclimatic
sequences on their own timescales. (a) Total arboreal pollen (AP %) and deciduous *Quercus* curve from
Lake Van (this study); (b) Arboreal pollen percentages from Yammoûneh basin (Lebanon; Gasse et al.,
2015); (c) AP including (green) and excluding (light green) *Pinus* and *Juniperus* (PJ) percentages of the
Tenaghi Philippon record (NE Greece; Tzedakis et al., 2003b); (d) AP sequence from Ioannina basin
including (orange) and excluding (light orange) *Pinus*, *Juniperus*, and *Betula* (PJB) (NW Greece;
Roucoux et al., 2011, 2008); (e) Lake Ohrid pollen record (AP %; Macedonia, Albania; Sadori et al.,
2016); (f) Stable oxygen isotope record of Lake Van ($\delta^{18}O_{bulk}$ data including the already published isotope
record of Kwiecien et al., 2014); (g) Peqi'in and Soreq Cave speleothem records (Israel; M. Bar-Matthews
& A. Ayalon, unpubl. data); (h) Synthetic Greenland ice-core record ($GL_{T-syn}$; Barker et al., 2011); (i)
Atmospheric $CO_2$ concentration from Vostok ice core, Antarctica (Petit et al., 1999); (j) Mid-June and
Mid-January insolation for 40°N (Berger, 1978; Berger et al., 2007). Bands highlights periods of
distinctive climate signature discussed in the text. Black dots mark significant interstadial periods. Marine
Isotope Stages is also shown (MIS; Lisiecki and Raymo, 2004). Termination III at 250 ka, TIIIA at 223 ka
and TII at 136 ka after Barker et al. (2011) and Stockhecke et al. (2014a).
**Fig. 6:** Comparison of the (a) last glacial period (MIS 4-2; Pickarski et al., 2015b) with the (b) penultimate
glacial (this study) characteristics at Lake Van. Shown is the insolation values (40°N, Wm$^{-2}$) after Berger
(1978) and Berger et al. (2007), the $\delta^{18}O$ profile from NGRIP ice core (Greenland; NGRIP members,
2004) labeled with Dansgaard-Oeschger (DO) events 1 to 19 for the last glacial period, the $\delta^{18}O$
composition of benthic foraminifera of the marine core MD01-2444 (Portuguese margin; Margari et al.,
2010) for the penultimate glacial, and the Lake Van paleovegetation with AP % (shown in black), AP in
10-fold exaggeration (grey line), Poaceae, deciduous *Quercus*, and *Pinus*. The grey boxes mark the
comparison between the different paleoenvironmental records of pronounced interstadial oscillations.
Marine Isotope Stage (MIS; Lisiecki and Raymo, 2004) and informally numbered interstadials of the
MD01-2444 record are indicated (Margari et al., 2010).

 **Tables:**

**Table 1:** Present-day climate data at Lake Van (see Fig. 1 for the location). Data were provided by the
Turkish State Meteorological Service (observation period: 1975-2008.
**Table 2:** Main palynological characteristics of the Lake Van pollen assemblage superzones (PAS) and
zones (PAZ) with composite depth (mcblf), age (ka BP), criteria for lower boundary, components of the
pollen assemblage (AP: arboreal pollen, NAP: non-arboreal pollen), green algae concentration (GA: low
<1000; high >1000 coenobia cm$^{-3}$), dinoflagellates concentrations (DC: low <100; high >100 cysts cm$^{-3}$),
charcoal concentrations (CC: low <2000; moderate 2000-4000; high >4000 particles cm$^{-3}$) and their
inferred dominated vegetation type during the penultimate interglacial-glacial cycle. Marine Isotope
Stages (MIS) after Lisiecki and Raymo (2004) were shown on the right.
**Figure 1:**

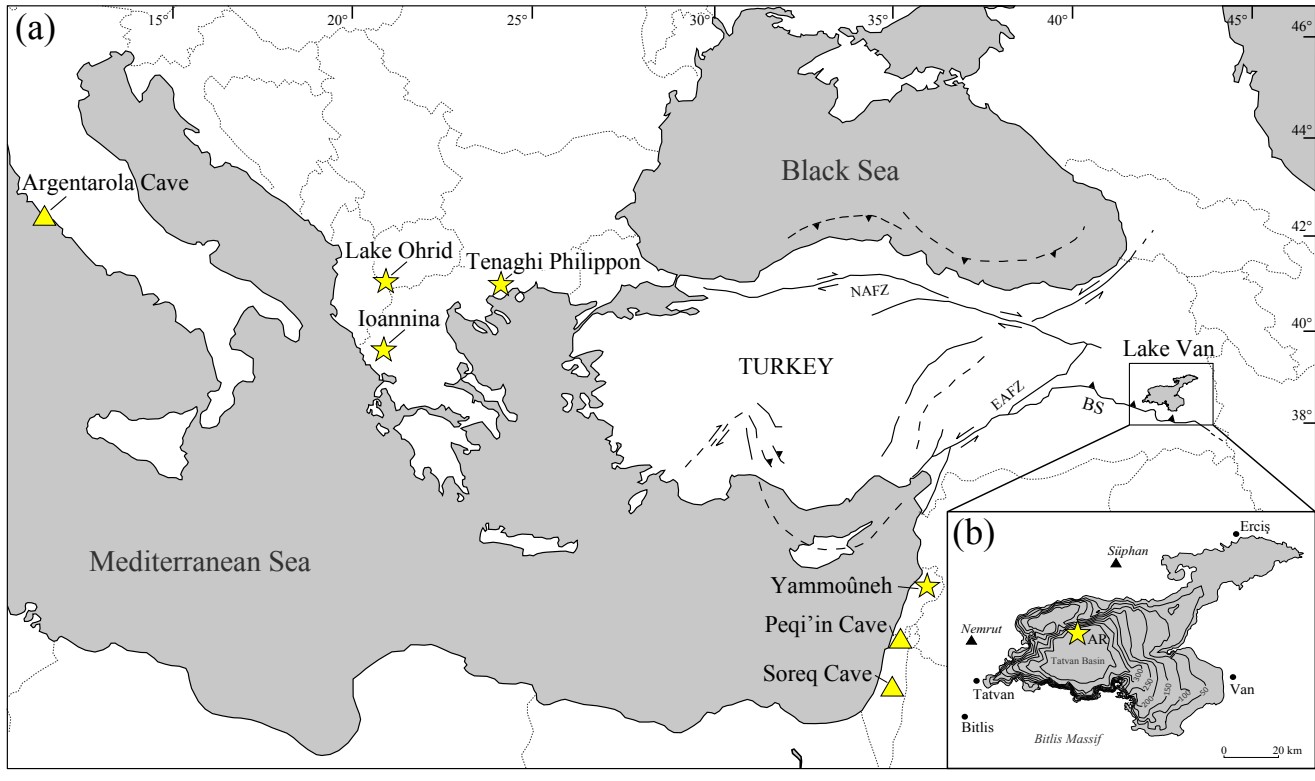

**Figure 2:**

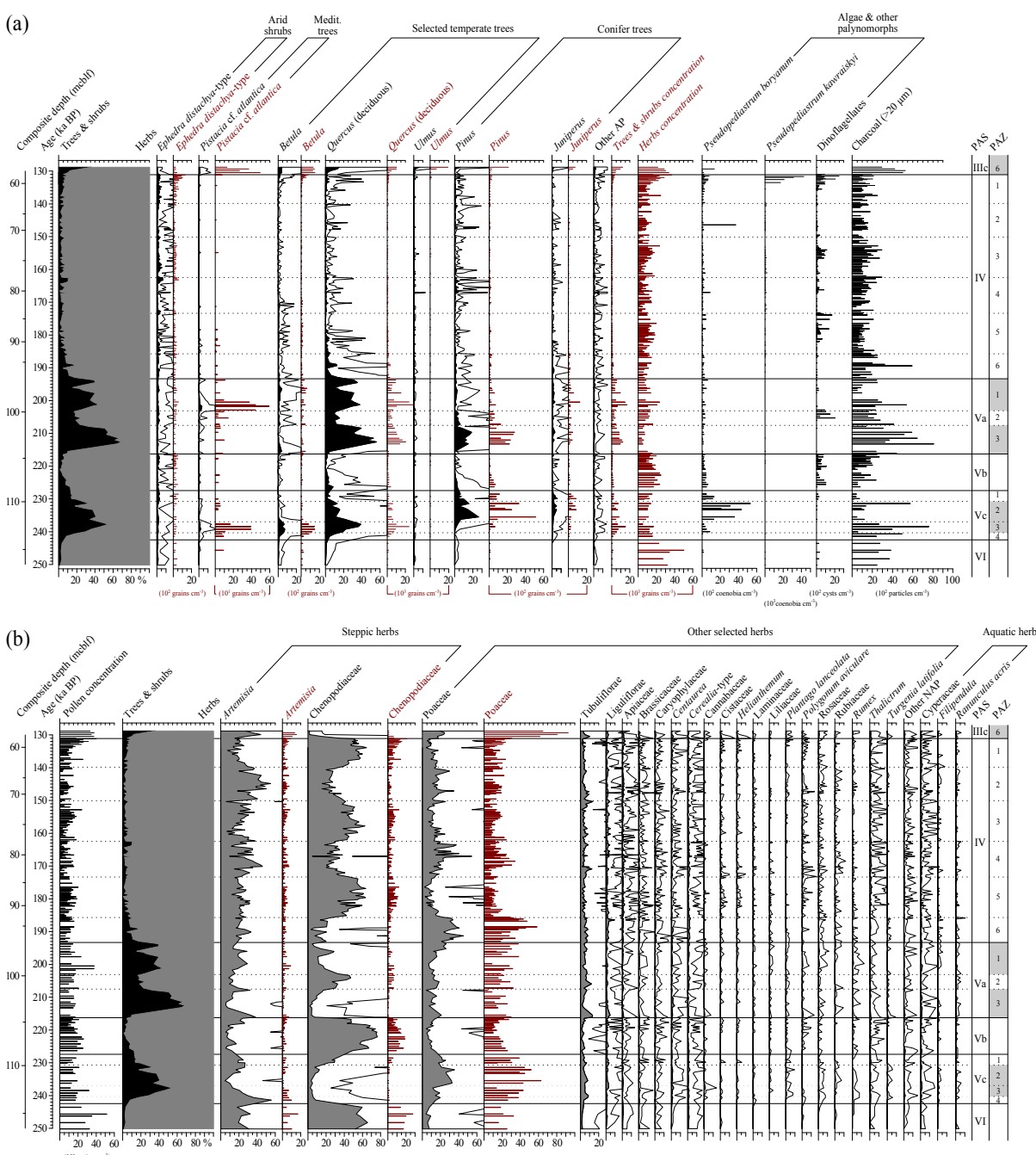

**Figure 3:**

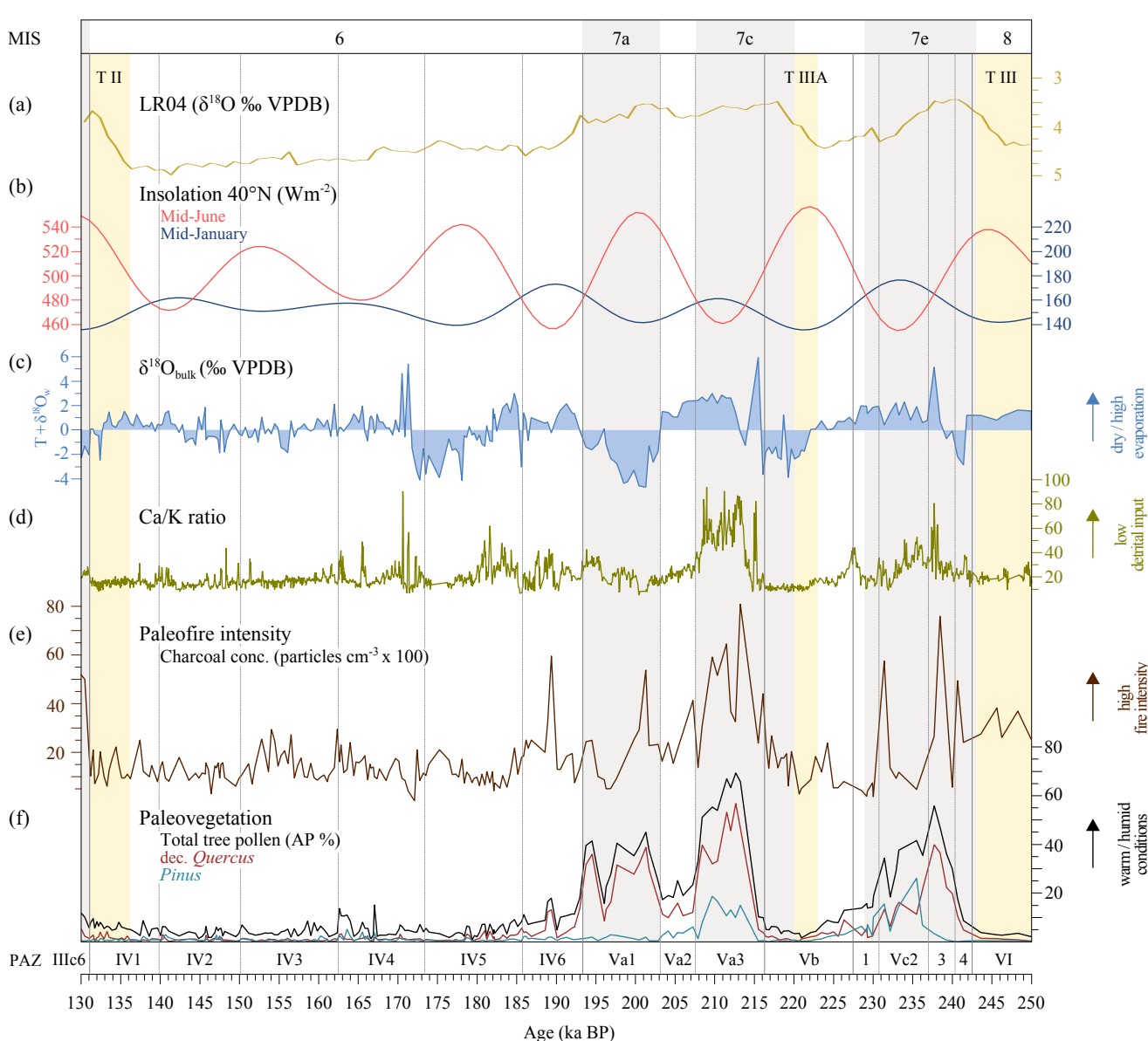

**Figure 4:**

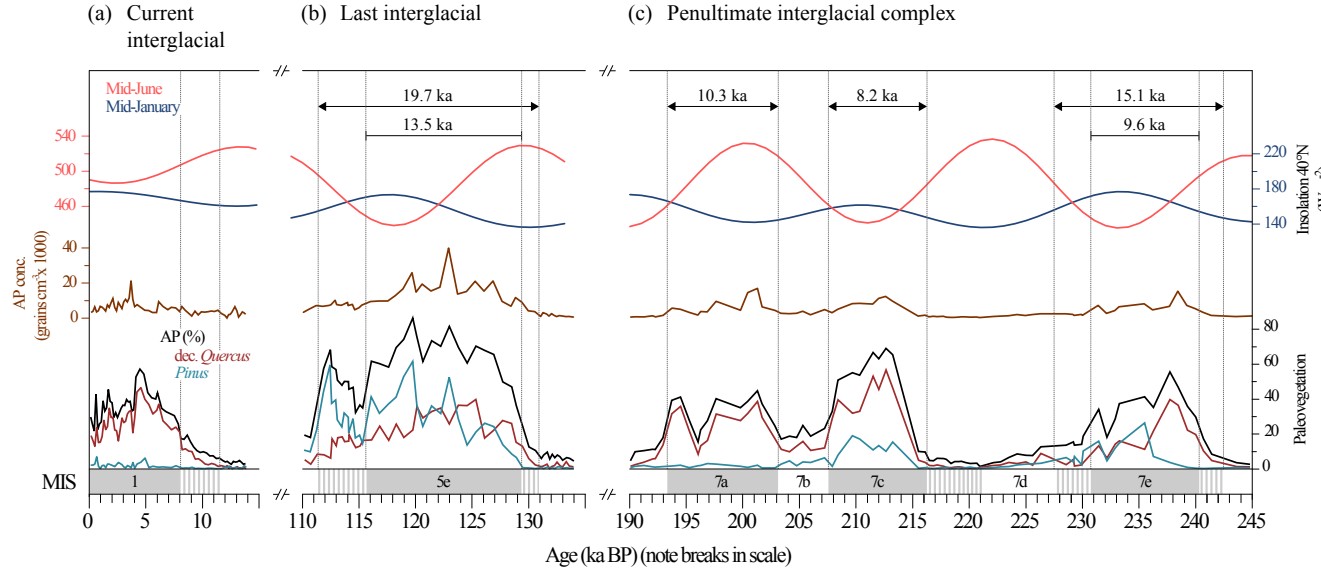

Age (ka BP) (note breaks in scale)

**Figure 5:**

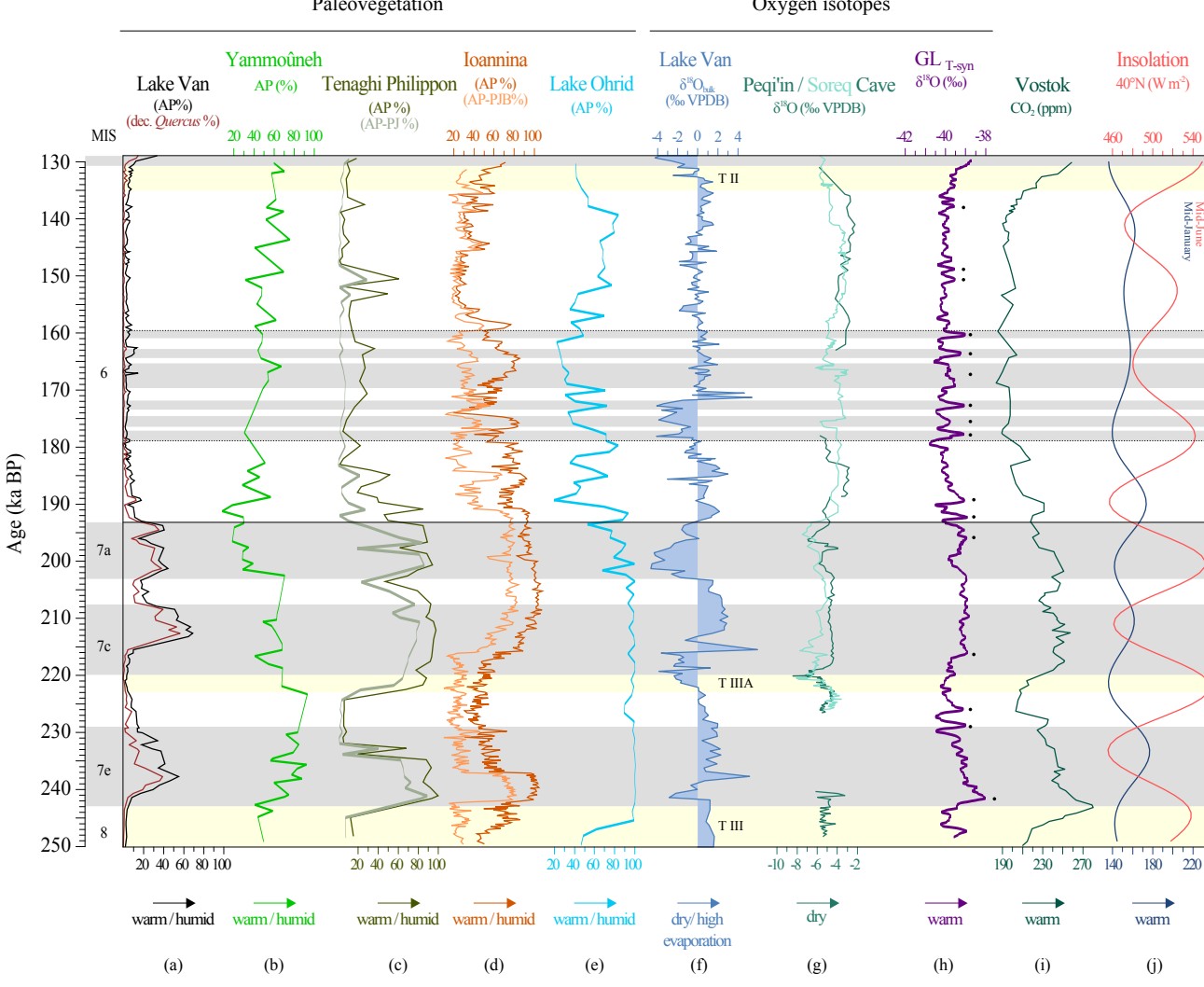

**Figure 6:**

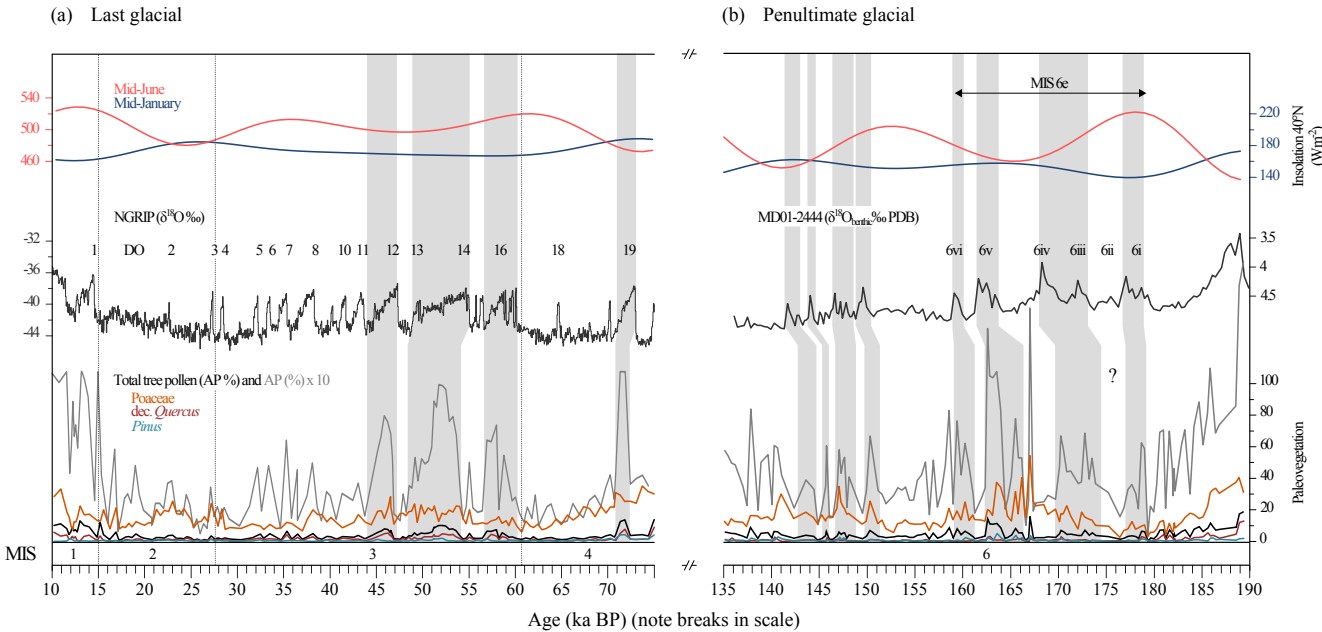

Age (ka BP) (note breaks in scale)

**Table 1:**

| Station | Coordinates | | | Mean temperature (°C) | | | Mean precipitation (mm) | | |
|---|---|---|---|---|---|---|---|---|---|
| | Latitude (°N) | Longitude (°E) | Altitude (m asl) | Jan | July | Year | Jan | July | Year |
| Bitlis | 38°24' | 42°06' | 1551 | -2.0 | 22.0 | 9.4 | 161 | 5 | 1232 |
| Tatvan | 38°30' | 42°17' | 1690 | -3.2 | 21.9 | 8.7 | 95 | 7 | 816 |
| Erciş | 39°20' | 43°22' | 1750 | -6.0 | 21.8 | 7.7 | 31 | 7 | 421 |
| Van | 38°27' | 43°19' | 1661 | -4.0 | 22.2 | 9.0 | 35 | 4 | 385 |



**Table 2:**

| PAS | PAZ | Composite depth (mcblf) | Age (ka BP) | Criteria for lower boundary | Main palynological characteristics (minimum – maximum in %) | Dominant vegetation type | MIS |
|---|---|---|---|---|---|---|---|
| IIIc | 6 | 57.10 - 58.09 | 128.8 – 131.21 | Occurrence *Pistacia* | **AP:** *Betula* (2-4%), dec. *Quercus* (1-13%), *Ephedra distachya*-type (0-3%), *Ulmus* (0-2%), *Juniperus* (0-1%), *Pinus* (0-1%), *Pistacia* cf. *atlantica* (0-1%)<br>**NAP:** *Artemisia* (16-49%), Poaceae (7-25%), Chenopodiaceae (2-52%)<br>**GA:** Low **DC:** Low **CC:** Moderate to high | Steppe taxa become less widespread, giving way to open grassland | 5e |
| IV | 1 | 58.09 - 63.25 | 131.21 - 139.87 | Chenopodiaceae >40% | **AP:** Low AP (2-8%); increased frequencies of *Ephedra distachya*-type (1-5%); dec. *Quercus*, *Betula*, *Pinus*, and *Juniperus* are abundant at low level<br>**NAP:** Chenopodiaceae (39-64%) show high values at the top, while *Artemisia* (8-29%) abundances decline; moderate Poaceae percentages<br>**GA:** Low **DC:** Low **CC:** Low to moderate | Open desert steppe vegetation | 6 |
| | 2 | 63.25 - 71.50 | 139.87 - 150.14 | Chenopodiaceae <40% | **AP:** Low AP (1-7%); temperate trees are present at low level<br>**NAP:** Expansion of *Artemisia* continues and peaks in the middle of the zone (54%); Chenopodiaceae percentages drop to 15-41%; moderate Poaceae values (11-34%)<br>**GA:** Low with a single peak at 146.4 ka (c. 3,700 coenobia cm$^{-3}$) **DC:** Low **CC:** Low | Productive dwarf shrub steppe vegetation | |
| | 3 | 71.50 - 77.72 | 150.14 - 162.49 | Chenopodiaceae >40%; decrease *Quercus* | **AP:** Dec. *Quercus*, *Betula*, *Pinus*, and *Juniperus* are continuously present at low level (AP 2-8%); increase of *Ephedra distachya*-type (1-6%)<br>**NAP:** Predominance of Chenopodiaceae (33-62%); *Artemisia* (6-38%) shows moderate values with increasing trend towards the top, Poaceae continuously present at ~13%<br>**GA:** High to low at the end of the zone **DC:** Low to high **CC:** Low to moderate | Open desert steppe vegetation | |
| | 4 | 77.72 - 83.84 | 162.49 - 173.38 | Chenopodiaceae <40%; increase *Quercus* | **AP:** Low AP (1-14%); moderate dec. *Quercus* (0-3%); decrease of *Betula* (0-2%), while *Pinus* (0-5%) and *Juniperus* (0-1%) percentages increase towards at the top<br>**NAP:** Predominance of *Artemisia* (10-46%) and Poaceae (8-54%); Chenopodiaceae abundances (5-40%) are reduced<br>**GA:** Low to high **DC:** Low **CC:** Low with moderate peaks | Fluctuation between open desert-steppe and grassland scattered with temperate trees | |
| | 5 | 83.84 - 93.51 | 173.38 - 185.74 | Chenopodiaceae >40% | **AP:** AP (1-9%) decrease continuously throughout the zone; mainly by dec. *Quercus* (0-4%)<br>**NAP:** Base marked by a pronounced expansion of Chenopodiaceae (33-64%); *Artemisia* continues from previous zone with max. 32%, while Poaceae decrease (3-18%) | Change from grassland to desert steppe vegetation at the end of the zone | |

| PAS | PAZ | Composite depth (mcblf) | Age (ka BP) | Criteria for lower boundary | Main palynological characteristics (minimum – maximum in %) | Dominant vegetation type | MIS |
|---|---|---|---|---|---|---|---|
| | 6 | 93.51 - 97.02 | 185.74 - 193.36 | Decrease *Quercus*; increase Poaceae | **GA:** Low **DC:** Low to high towards the top **CC:** Low<br>**AP:** Reduction of AP; still abundant: dec. *Quercus* (1-31%), *Betula* (0-2%), and *Ulmus* (<1%); moderate conifer trees with small oscillations; disappearance of *Pistacia* cf. *atlantica*<br>**NAP:** Increase of Poaceae (21-45%); steppic herbs continue to be moderate<br>**GA:** Low **DC:** Low **CC:** Low to moderate, peak at 189.4 ka | Open grasslands with scattered temperate trees | |
| Va | 1 | 97.02 - 99.88 | 193.36 - 203.11 | Increase AP; peak *Pistacia* | **AP:** High AP (24-44%), e.g., dec. *Quercus* (8-38%), increasing values of *Betula* (0-4%), *Pinus* (0-3%), and *Juniperus* (0-3%); peak of *Pistacia* cf. *atlantica* (c. 3%) at the beginning; high tree concentration (>3,000 grains cm$^{-3}$)<br>**NAP:** Moderate percentages of steppic herbs (*Artemisia* 13-29% and Chenopodiaceae 11-33%) with significant peak of NAP (85%) near the base<br>**GA:** Low **DC:** Low **CC:** Low to moderate with one single high peak at 201.3 ka (>5,000 particles cm$^{-3}$) | Expansion of oak steppe-forest along with Mediterranean taxa (*Pistacia*), short-term influence of steppe vegetation | 7a |
| | 2 | 99.88 - 101.30 | 203.11 - 207.56 | AP <40%; decrease *Quercus* | **AP:** Reduced AP values (17-50%) mainly by dec. *Quercus* (10-30%) and *Pinus* (1-8%) but still above 15%; increase of *Ephedra distachya*-type (1-3%) and *Betula* (0-2%)<br>**NAP:** Expansion of Chenopodiaceae (15-47%), peak of *Artemisia* (9-32%) at the beginning; moderate Poaceae (5-19%)<br>**GA:** Low **DC:** Low to high **CC:** Low to moderate | More open (steppe) landscape with still patchy pioneer & temperate tree | 7b |
| | 3 | 101.30 - 104.19 | 207.56 - 216.28 | Chenopodiaceae <40%; increase *Quercus* | **AP:** Predominance of dec. *Quercus* (2-56%) with significant peak at 102.8 mcblf (212.6 ka) followed by a decreasing trend; high values of *Pinus* (0-19%); *Betula* (0-4%) and *Juniperus* (0-2%) are abundant; *Pistacia* cf. *atlantica* and *Ulmus* pollen occur sporadically; high AP concentration (>3,000 grains cm$^{-3}$)<br>**NAP:** Peak of *Artemisia* (6-38%), Poaceae (5-21%), and Tubuliflorae (2-13%) at the beginning; very low Chenopodiaceae values (4-48%)<br>**GA:** Low **DC:** No occurrence **CC:** High | Expansion of oak-pine steppe-forest | 7c |
| Vb | | 104.19 - 109.05 | 216.28 - 227.42 | Chenopodiaceae >40% | **AP:** Very low AP percentages (1-12%) and concentration (<2,000 grains cm$^{-3}$); decrease of dec. *Quercus* (0-9%), *Pinus* (0-3%), and *Juniperus* (<1%)<br>**NAP:** Predominance of Chenopodiaceae (37-76%); Poaceae (4-15%), and *Artemisia* (6-26%) are abundant<br>**GA:** Low **DC:** Low **CC:** Low with moderate values at the end | Extensive desert steppe vegetation | 7d |

| PAS | PAZ | Composite depth (mcblf) | Age (ka BP) | Criteria for lower boundary | Main palynological characteristics (minimum – maximum in %) | Dominant vegetation type | MIS |
|---|---|---|---|---|---|---|---|
| Vc | 1 | 109.05 - 109.94 | 227.42 - 230.71 | Disappearance *Pistacia*; decrease AP, increase Chenopodiaceae | **AP:** Decrease in AP (14-19%), mainly dec. *Quercus* (2-5%), *Pinus* (2-10%); *Pistacia* cf. *atlantica* disappears<br>**NAP:** Strong increase in Chenopodiaceae (23-32%), reduced *Artemisia* (19-27 %) and Poaceae (18-26%)<br>**GA:** Low **DC:** Low **CC:** Low | Increasing influence of steppe taxa, expansion of open vegetation | 7e |
| | 2 | 109.94 - 111.73 | 230.71 - 236.95 | Decrease *Quercus* and *Pistacia*; increase *Pinus* | **AP:** Percentages of dec. *Quercus* (6-21%), *Betula* (0-1% and *Pistacia* cf. *atlantica* decline while those of *Pinus* (4-26%) and *Juniperus* (2-5%) rise<br>**NAP:** Increased steppic taxa, e.g., *Artemisia* (5-26%) and Poaceae (21-36%); still low Chenopodiaceae (3-13%)<br>**GA:** High **DC:** Low **CC:** Low with one peak at the end | All temperate tree taxa declined gradually, while *Pinus* and grassland expanded (Pinus-dominated steppe-forest) | |
| | 3 | 111.73 - 112.64 | 236.95 - 240.31 | *Quercus* >10%; Chenopodiaceae <40% | **AP:** Peak values for *Betula* (4-8%) and *Pistacia* cf. *atlantica* (1-2%), expansion of dec. *Quercus* (10-40%); *Pinus* (0-3%), *Juniperus* (0-1%), and *Ulmus* are abundant; highest AP concentration (c. 5,300-15,300 grains $cm^{-3}$)<br>**NAP:** Retreat in steppe percentages mainly *Artemisia* (13-37%) Chenopodiaceae (3-6%); moderate Poaceae values (12-20%)<br>**GA:** Low **DC:** No occurrence **CC:** Moderate to high | Expansion of oak steppe-forest along with Mediterranean sclerophylls (*Pistacia*) | |
| | 4 | 112.64 - 113.70 | 240.31- 242.48 | Occurrence *Pistacia* | **AP:** Increase in temperate AP, e.g., dec. *Quercus* (1-10%) and *Betula* (1-5%); occurrence of *Pistacia* cf. *atlantica* (~1%), *Juniperus* (~1%), and *Ulmus* (sporadic)<br>**NAP:** Herbaceous taxa continue, mainly Poaceae (7-20%) and *Artemisia* (37-56%); Chenopodiaceae decrease (6-59%)<br>**GA:** Low **DC:** No occurrence **CC:** Moderate to high | Steppe taxa become less widespread, giving way to open grassland | |
| VI | | 113.70 - 117.19 | 242.48 - 250.16 | Not defined | **AP:** Very low abundances of AP (*Betula* 0-1% and dec. *Quercus* 0-1%), very low tree concentration (c. 570-1320 grains $cm^{-3}$)<br>**NAP:** Predominance of steppe taxa, mainly Chenopodiaceae (52-66%) and *Artemisia* (18-33%)<br>**GA:** Low **DC:** Low **CC:** Moderate | Extensive open desert-steppe vegetation | 8 |
