# Peer review of "A new high-resolution pollen sequence at Lake Van, Turkey: Insights into penultimate interglacial- glacial climate change on vegetation history"

_Climate of the Past, 2016_

## Referee Comment (RC1)

Thank you for the opportunity to comment on this manuscript. It concerns a sound data set of great value to the palaeoecological community as it comes from a region where such data for this period are scarce. Overall, I think the work is good and should be published in CPD but there are a number of important details that need to be considered and corrected first. In many instances these are related to terminology, definition of terms and ambiguity or circularity in the phrasing. One important example of this is the use of the term "steppe forest" without definition or explanation. Another is the use of marine isotope stage names to refer directly to intervals identified in the pollen record with no explanation for how that equivalence was established (even once an explanation is given, MIS terms should not be used directly for terrestrial intervals - see further details below). There is occasional circularity and a lack of clarity as to what was used to infer what but this is usually a question of the phrasing, and not a fundamental problem with the argument (examples below). I think that to make a convincing argument, the basis of both the stratigraphy and the chronology should be outlined in more detail (even if they are described elsewhere) so that the paper can stand alone. Without this, it is difficult to assess the validity of statements about the relative timing of events in the Lake Van pollen record and global scale climatic events. The vegetation reconstructions/inferences (particularly those involving trees) I think need to be more clearly described and the basis for the inferences better founded (e.g. with reference to modern pollen-vegetation studies, where possible). Also, I think there IS succession where the authors have suggested there is not... this could do with some more consideration.

These issues are detailed below along with suggestions for grammatical corrections. In addition there are numerous minor grammatical errors (especially plural/singular, tenses) (not all are listed below).

Comments

| Page | Line | Comment |
|---|---|---|
| | | |
| 1 | 11 | "effective moisture" needs some qualification (high, low?) otherwise the meaning is not clear |
| 1 | 12 | "forest" ought probably to be qualified with "open" since this is "steppe forest" |
| 1 | 12 | I think the conventional term for the biome is "wooded steppe" (e.g. Allen et al. 1999, Nature 400, 740 – 743). If "steppe forest" means something other than this, then it must be defined (and in any case, a reference is needed). |
| 1 | 13 | "The warmest stage…" The previous sentence suggests moisture is the main limiting factor. If temperature is important too, then need to make it clear that both are involved throughout the text (i.e. avoid summarising warmer/wetter as "warmer"). |
| 1 | 13 | "in terms of" I think this should read "as indicated by" |
| 1 | 13 | "amplitude" Double check – do you mean amplitude, or duration (or both)? Please clarify this. |
| 1 | 14 | Insert "… the tree population maximum associated with…" before "MIS 7" |
| 1 | 17 | Clarify presence or absence of trees in this instance of "steppe" |
| 1 | 19 | Replace "more" with "higher" |
| 1 | 21 | The mild conditions inferred here are also in agreement with pollen records from |

elsewhere in southern Europe.

| | | |
|---|---|---|
| 1 | 25 | Insert after "subdued oscillations": "… as in other records of this interval from southern Europe." E.g. MD01-2444 and I-284. |
| 1 | 27 | Clarify what it is that indicates cooler and wetter conditions (it's not the identification of MIS 6e!) |
| 2 | 36 | Could you say what the resolution was in that study? |
| 2 | 41 | Replace "allow" with "have allowed" |
| 2 | 45 | Replace "is not being" with "has not been" |
| 2 | 49 | Replace "already available" with "existing" |
| 2 | 57 | Replace "this presented study" with "our" |
| 2 | 57 | Delete "want to" |
| 2 | 58 | Change to past tense |
| 2 | 61 | Change to past tense |
| 3 | 67 | "meter" should be plural |
| 3 | 77 | "latitudes" should be singular |
| 3 | 88 | It would be helpful to know whether these forest and shrub formations represent the "natural" state of the vegetation versus the result of human impacts (e.g. pastoralism). |
| 4 | 99 | "those" is ambiguous… can you say what "those" refers to? (Existing pollen data?) |
| 4 | 101 | Chronology section – perhaps the explanation of how (at least this part of) the Lake Van sequence has been aligned to the marine isotope stratigraphy belongs here? |
| 4 | 114 | How were the age control points identified – in which proxy record? |
| 3 | 127 | Insert "group" after "taxonomic" |
| 5 | 131 | "percentages" should be singular |
| 5 | 133 | Should be "lake surface"? |
| 5 | 137 | Replace "was" with "were" |
| 5 | 142 | Insert "were made" after "measurements" |
| 5 | 157 | "deciduous forested"… I think you need to specify whether this is closed or open forest because the implication of following this with "open steppic landscapes" is that the forest was closed canopy which, given the low AP%, is unlikely. Which leads to the next comment… |
| 5 | 159 | With low AP % values, I'm not sure "forested" gives the right impression. It sounds a bit too, well, "forested"! Is there an alternative term that would be a better representation of the open landscape with few trees that the pollen data seem to represent? Ideally, this would have its basis in modern pollen-vegetation work. |

| 6 | 166-7 | "The highest concentration peaks occur during forest intervals". Please rephrase this to remove the circularity. (How do we know these were forested intervals? Partly, because of the high pollen concentration!). |
| --- | --- | --- |
| 6 | 171 | Add a brief comparison with the pollen record here (to be consistent with the next sentences about Pediastrum which are compared with the pollen). |
| 6 | 181 | Is the amplitude exceptionally high? This phrasing suggests you have made comparisons with other records… if so, please indicate broadly which records (or kinds of records) it is high relative to. |
| 7 | 197 | Please say how you define stadial and interstadial here OR avoid using these terms here and make the correlation between particular peaks in the isotope curve and particular stadial-interstadial transitions (defined in other records) later on. I think the same applies for Termination III (since you haven't yet clearly justified the identification of TIII in the lake Van record). |
| 7 | 204 | This should read "marked" not "remarkable" |
| 7 | 205 | Does "here" refer to this study? If so, the "generally considered" does not make sense. Please clarify. |
| 7 | 206 | The sentence starting "This general pattern…" is ungrammatical. The warm phases alternate with the cold phases. Please rephrase. Also, it is interesting that you do not mention changes in moisture availability here. There either needs to be a justification for that (deliberate?) omission, or both climatic parameters need to be considered. |
| 7 | 207 | There is something odd about the line of reasoning here. On what basis did you establish the equivalence between the phases with more trees and MIS 7e, 7c and 7a if not by comparison of the pollen record with a marine isotope record (such as that used in the stratigraphy of Martinson et al. 1987) directly or indirectly? (I.e. "comparable with the marine classification by Martinson…" does not make sense). Also, take care with using the language of marine isotope stratigraphy to directly refer to intervals recorded by the pollen record – it is not strictly correct to do that (though of course we all do it informally). Ideally, wooded intervals in the AP% curve should not be directly aligned with the (apparently) equivalent MIS stages; there are significant offsets (and uncertainties) in the timing of the beginning and end of forest intervals on land relative to the beginning and end of warm marine isotope stages. Marine pollen records, form the Iberian margin and elsewhere, which combine marine isotope stratigraphy with a terrestrial vegetation signal are the only records in which the relative timing can be established directly (and these show significant offsets). |
| 7 | 214 | Should "abrupt" be "brief"? In this context, "abrupt" doesn't really make sense. |
| 7 | 217 | Rephrase: link the sentence starting "It is clear…" to the previous one and remove "it", which is unclear. |
| 7 | 217 | "… vegetation communities changed." State what kind/direction of changes this refers to. |
| 7 | 219 | When discussing events in the past, not stratigraphy, the terminology is "start" not "base" |
| 7 | 222-3 | The inference of "oak steppe-forests where summer-green Quercus rises consistently above 20%" needs a few words of explanation and justification, and a reference. |
| 8 | 233 | "this" should be "their" as in "their hypothesis", but it would also be helpful to have a brief re-statement of that hypothesis here. |

| 8 | 241 | I think there needs to be a clear statement of how these records were aligned – i.e. how do you know that the vegetation changes (that you interpret to represent cooling/drying) recorded in the Lake Van sequences occurred BEFORE "… ice accumulation is evident… in MD01… "? |
|---|---|---|
| 8 | 241 | The linking phrase ("In light of these insights…") does not work because the insights just described are not what suggests a shift from temperate to coniferous taxa. |
| 8 | 248 | Why "re-expansion" not just "expansion" (implies a second expansion)? |
| 8 | 253 | The persistence of relatively large tree populations through the period equivalent to MIS 7b was noted at Lac du Bouchet and at Ioannina; please cite this work here. |
| 8 | 263 | MIS 7c is not an interstadial… unless you want to define it as such at Lake Van (but then this must be explained and justified). |
| 9 | 263-266 | All good reasons listed here for not calling MIS 7c an interstadial. |
| 9 | 273 | Which other tree taxa are missing, besides Pistacia, from the succession… I couldn't see any others. If only Pistacia is missing from the wooded interval equivalent to MIS 7c, this is not sufficient to say there is no succession. I think there is: as in the "7e" interval the "7c" tree population expansions begin with Betula, continue with Quercus and this is followed by expansion of Pinus populations. |
| 9 | 275 | Ensure the phrasing reflects the fact that you are describing conditions in the region of lake Van and that the same conclusions may not apply elsewhere (i.e. include reference to the region to which your conclusion applies). |
| 9 | 277 | Don't need BP with ka, conventionally. |
| 9 | 277 | Along with the intervals that have more trees, the open (treeless) intervals also need to have their equivalence to the marine isotope stratigraphy justified. To repeat – it is not good practice to refer directly to intervals identified in a terrestrial pollen record with the MIS nomenclature (you need to demonstrate the basis for the correlation, and even then, I would still say "the interval broadly equivalent to MIS…" or similar wording). |
| 9 | 277 | Related to the comment above, replace "MIS 7d" with "pollen record between … and … ka" or use zone names. There are numerous other places in the manuscript where MIS terminology is used where it is not appropriate. |
| 9 | 290 | Please give references (after "… Lebanon and southern Europe.") |
| 9 | 293-4 | This description of the vegetation during the interval equivalent to MIS 7b is not consistent with the description of this interval above (where 7d and 7b, to use the informal shorthand, are described together as having "extensive steppe vegetation… [and] inhibited tree growth…" |
| 9 | 297 | Why is higher in '…'? (another occurrence in line 304: 'high') |
| 10 | 299 | Delete "arboreal" |
| 10 | 300 | "i.e." should be "e.g." here |
| 10 | 305 | Check – if $CO_2$ was higher in 7b, it is more likely to have been warmer than 7d. |
| 10 | 308 and onwar | Consider using past tense in this section as it discusses events in the past rather than the record of those events. |

| | | |
|---|---|---|
| | ds | |
| 10 | 315 | Delay relative to what…? |
| 10 | 318 | Replace "due" with "indicated by" |
| 10 | 324 | "However…" doesn't make sense here. |
| 10 | 327 | Reference required (to support observation about range of ecological requirements within the Quercus genus). |
| 10 | 328 | There seem to be some logical steps missing… can this be explained more clearly? Make clear that both abundance and composition of tree populations differs. Also, it is necessary to reconcile this argument for wetter/cooler conditions with the presence of Pistacia close to the start of the "forest" interval corresponding to MIS 7e. |
| 11 | 335-6 | This assumes that the "climate optimum" is equivalent to the "terrestrial temperate interval" – either justify this equivalents or use "terrestrial temperate interval" both times. |
| 11 | 341 | Replace "evident" with "suggested" or "indicated" |
| 11 | 343 | The "rapid decline in temperate trees" does not make sense… which decline does this refer to? |
| 11 | 350-351 | "… resembles the pattern of interstadial to stadial stages." - as defined by what? |
| 11 | 359 | A landscape cannot be "less extensive" |
| 11 | 360 | "greater" would make more sense that "great" here. |
| 11 | 362 | Replace "values" with "populations" as this is about inferred vegetation now, rather than the pollen record. |
| 12 | 370 | Should also cite Margari et al. 2010 for the Iberian margin marine pollen record of MIS 6. |
| 12 | 379 | … and this pattern is also recorded at Lake Van? |
| 12 | 385 | Which transition does this refer to? Or should it read "transitions"? |
| 12 | 394 | At face value, this should not be the only reference given for the DO events 17 to 12. |
| 12 | 396 | Not clear what is meant by "compared to"… do you mean "comparable with" or "similar to"? Or something else? |
| 12 | 397 | "Intensities" is ambiguous… should it be amplitude? |
| 13 | 403 | "is supported by" should be "suggests" if the pollen forms the basis of this climatic inference. |
| 13 | 409 | "… points to a general picture of cold but 'wet' conditions during MIS 6e than experienced during MIS 3." This is not grammatically correct. |
| 13 | 426-7 | It is not clear to me what this vegetation formation would look like. Which aspect was "dense"? I think the term "steppe" is incompatible with the term "dense forest" unless the two kinds of vegetation occurred simultaneously in different areas (e.g. open steppes with discrete areas of dense forest… but that wouldn't be called a "steppe forest") |

| | | |
|---|---|---|
| 13-14 | 431-2 | "…strong thermal and hydrological seasonal contrasts during the last interglacial, and a higher humidity during the Holocene climate optimum…" are not discussed in the rest of the manuscript. If they are to appear in the conclusions, they need to appear earlier in the text as well. |
| 15 | 453 | Check spelling of Miriam… |
| 20 | Fig 2 | Please add an indication of the basis on which the taxa shown were selected (ecological importance, abundant… ?). It would be helpful to know how many AP taxa are not included (and what proportion of the sum this represents). A curve for "other AP" would demonstrate this (if they are too rare to show, then this needs to be said). The same point applies to the NAP. |
| | | Also, the curves are black (the fill is white). |
| 21 | Fig 3 | Please indicate on what basis the MIS equivalents are assigned. Even if this is addressed in another paper, for this paper to make a convincing case, it needs to be said here too. |
| 22 | Fig 4 | Add a statement to explain on what basis the interglacials illustrated here (MIS 5e, 7e) are defined (because under some definitions, 7c could also be an interglacial). |
| 23 | Fig 5 | Inclusion of AP-PJB% from Ioannina (as well as AP%) would have been more informative as this signal is more sensitive to climatic fluctuation and picks out a very similar pattern to that in lake Van… e.g. the minor decline of temperate tree populations associated with MIS 7b and a post-MIS 7a millennial scale oscillation. |
| | | Caption: Is it a "correlation scheme" if each curve is presented on its own timescale? There are some pronounced offsets in the timing of major vegetation changes which seem too large to be real and are likely to be exaggerated by age uncertainties. Could you clarify how this diagram was constructed (in the caption if not in the text), where timescales align NECESSARILY (because of the way the age models have been developed, for example) and where timescales are the original published ones (and the sources for those age models… for example, have you placed the I-284 curve on the GL synth timescale, or on the timescale published in 2008?). Without this kind of information, it is difficult for the reader to understand the significance of apparent alignments and offsets. |

---

## Short Comment (SC1) · 19 Jan 2017

General comment:

This is a very interesting work, clearly presented and well discussed. It provides new detailed information on the vegetation development and climate changes in the eastern Mediterranean region in the time interval 250-130 ka.

My main concern relates to the succession of vegetation types claimed for the warm stages in the paragraphs "forested periods" and "conclusions". At lines 212-214 it is stated that "the vegetation succession starts with the colonization of open habitats by

pioneer trees, such as Betula, followed by sclerophyllous Pistacia cf. atlantica and a gradual expansion of deciduous Quercus". "The ensuing ecological succession at Lake Van is documented by high percentages of dry-tolerant and/or cold-adapted coniferous species (lines 235-236). This succession is not clearly visible in any of the forest phases: deciduous Quercus is always starting before or at the same time of Betula. Pistacia is almost missing in the forest phase corresponding to MIS 7 c. Pinus develops after Quercus in MIS 7e, but it is coeval in MIS 7c and is very sparse in MIS 7a. I suggest to avoid this generalization, which may be of interest in central Europe, far away from glacial refugia, but is not appropriate for the Lake Van region. Similarly, I do not find in the diagrams the most depleted (negative) delta18Obulk values at the base of each early temperate stage (lines 218-219): this is true at the onset of MIS 7e, but the next time interval with low values of delta18O is found during the cold stage of MIS 7d, and the following low values are recorded during the warm stage of MIS 7a. Thus, the suggested generalization is not convincing.

Minor corrections:

Line 55: Roucoux et al., 2008, 2011

Line 56: Tzedakis et al., 2003b, 2006

Line 79: there is no correspondence of rainfall values with table I. Besides, Van is not in the north-east (Erciş is in the NE)

Line 82: the vegetation cover around Lake Van

Line 87: dominated by dwarf-shrub steppes

Line 122: HCl

Line 131: diagram of selected taxa. You may consider to add here that the complete list of data will be available on PANGEA

Line 139: on bulk sediment samples

[Figure]

Line 163: (max. ∼70 %)

Lines 212-213 and 235-237: see general comments

Line 236: that suggest a cooling

Line 240: in other words. . . . . .this sentence seems incomplete

Line 250: from a variety of factors

Line 274: species

Line 280: tolerates

Line 347: Between 193 and 157 ka BP

Line 350: an age of ∼189 ka BP is probably too precise with respect to the chronological setting of the record. Better say ∼189 ka BP

Line 371: shares

Line 373: refers

Line 408: colder but wetter climate conditions during MIS 6e than during MIS 3

Line 422: a unique record

Lines 428-430: see general comments

Line 439: wetter conditions than during MIS 3

Lines 441-444: In the last sentence of the conclusions it would be better to emphasize what is new in your record instead of remarking what was already known

---

## Referee Comment (RC2) · Anonymous Referee #2 · 31 Jan 2017

General comments:

This is an exciting and valuable new data set and a major contribution to the knowledge of past climate and vegetation development in the Middle East.

As you clearly show in your discussion, vegetational development, i.e. transitions from open steppe vegetation to various stages of deciduous and coniferous woodland (and vice versa) are not only driven by temperature, but largely by moisture sources and availability. Thus it is somewhat risky to directly relate the Lake Van pollen and oxygen isotope records to the marine isotope stratigraphy and to use the MIS terminology. I

suggest to interpret the Lake Van record with regard to regional climate and vegetational change and use it as a basis for discussing possible correlations to the MIS, insolation etc.

Be careful with the term 'succession'; I think it may not be used in a central european sense. At Lake Van there is a distinct gradient in moisture from south-west to northeast; the 'succession' from open steppe to deciduous oak woodland as described here might rather be a movement of the different vegetation formations from SW to NE than an all-over woodland expansion.

Minor improvements and suggestions:

(Linguistic and grammatical improvements suggested by referee#1 are not repeted here)

Line 114: please provide some additional information on how you did the synchronization

Lines 157 ff.: you may replace 'forest' by 'woodland', '(sparsely) wooded landscape' ....

Line 163: Chenopodiaceae max. 70%

Line 217: how did the vegetation change?

Lines 231-234: please give a brief description of this relationship

Lines 239-241: What is the link from the Lake Van vegetation to the MD01-2447 record based on?

Line 315: delayed - relative to what?

Line 347: replace 'during' by 'between'

Line 379: Do you have any idea, why this evidence is missing at Lake Van?

Line 426: 'dense' does not really fit with a steppe-forest - maybe 'well developed'

Lines 441-444: Please add afew words saying what is different / new / special at Lake Van

Fig. 2b: Why is Thalictrum in the aquatic group? there are about 30 species in Antolia, most of them adapted to dry conditions, some prefer humid soils, but there is no real aquatic species.

---

## Referee Comment (RC3) · G. Jiménez Moreno (Referee) · 2 Feb 2017

This is an interesting article that shows new detailed pollen and oxygen iso-tope data from the MIS 8-6 part of the Lake Van sedimentary record. The authors interpret the pollen and isotope changes as changes in vegetation and precipitation/evapotranspiration around the lake basin. Vegetation changes be-tween forested-steppe environments can be correlated with climate oscillations (interglacial/interstadials-glacial/stadials) described in the marine isotope records. The paper is well-written and the data support the interpretations/conclusions and thus deserves publication in CP. However, in my opinion, there are some changes that need to be done before publication and several topics are not very well discussed in the manuscript and need to be clarified. Below are my comments:

-It is not very clear what is really triggering the vegetation changes in the area - is it mostly temperature or precipitation? In some parts of the text temperature is indicated as the main trigger and in some others is precipitation or effective precipitation (supported by the isotope data). A clear example is the Abstract (lines 11-13) where effective precipitation is first introduced as the main trigger and then temperature...and this is very confusing as maximum insolation and thus maximum temperature would reduce the effective precipitation and should not produce the same effect on the vegetation. For example, in line 13 – maximum forest development during stage 7c does not seem to occur during summer insolation maxima. . ...

In this area, where precipitation is not very abundant I would think that forest development would be mostly related to precipitation or effective precipitation. I think you should be consistent throughout the text.

-I also had the feeling that after reading the text and looking at the figures one still lacks of a clear idea of what is the relationship between insolation and plant dynamics in this record. In lines 268-269 it is stated that ". . .vegetation development (forest?) is clearly controlled by insolation forcing and associated climate regimes (high summer temperature, high winter precipitation)". I understand here that forest development in this area is "clearly" controlled by summer insolation, so in a very simplistic model if we had high summer insolation we would have had high forest development. This is a model that can be applied to several long Mediterranean records (see Tzedakis et al., 2007). However, if we look at figure 3, the major forest development seems to happen during summer insolation minima, so completely the opposite of what it is said in the text. Check stages 7e and 7c. What I understand from this is that forest cannot develop during periods of insolation maxima (and probably precipitation maxima) due to very high evaporation and that would explain the big lag between them. The vaguely

mentioned lag in the text (line 315) is not just 2-3 ka. . .but about 10 ka (ie. stage 7c). This subject should be further explained and clarified in the text.

-I am also puzzled about the isotope record from the lake and the comparison with the pollen data. First, if the interpretation of the data is correct (higher values, higher evaporation/dryness), the isotope data do not seem to agree with the summer insolation and it should. Second, if the vegetation was delayed because during summer insolation maxima there was too much evaporation, this would show a delay between the isotope data and the pollen and they basically covariate (except for some periods (stage 7a). Please clarify.

-The fact that stage 7c shows one of the largest forest development in the record needs to be highlighted in the chapter about "Comparison of past interglacials at Lake Van".

-It is very confusing to see terms such as "steppe forest landscape" "oak-pine steppe forest" or "oak steppe forest" as these two terms "forest" and "steppe" are quite opposite. Why not calling these forests with AP pollen percentages around 60% "forests" or if you do not agree that they are close forests "open forests"? Also, steppes are mostly characterized in the area by Artemisia and Amaranthaceae, and Poaceae seems to be relatively abundant during the "forest" periods so it would not be quite an "steppe" environment.

-Even though there is certain variability during MIS 6 the forest oscillations are only between 0-10%. I would not call these oscillations "pronounced" as stated in the abstract (line 23). The authors should soften the language regarding these oscillations (section 4.2).

-Line 10. "The presented record displays the highest temporal resolution for this interval"? from where? Lake Van? Turkey? The World? Please be specific.

-Pinus has an important role in the observed vegetation changes in this record and probably were important tree taxa regionally as well. Therefore, I think the authors

should give some information about Pinus distribution in the area or regionally at Present in "Site description" as later it is mentioned that was transported to the area by the wind (lines 235-238).

-Line 120. Give unit for the "4 cm" size samples - cm3 ?

-I think the presence of Spiniferites should be better explained as many people would interpret this taxa as marine species. Do they occur in lake environments? Under what circumstances?

-Lines 195-196. "The d18O composition of the lake water becomes progressively more enriched during interglacial/interstadial periods". Not fully true – check stage 7a where the opposite happened. Please be more specific.

-Lines 197-200 – This statement is not clear – in Fig. 3 the isotopic changes are explained and changes in dryness or evapotranspiration, supported by low detritic input in the lake.

-The charcoal record is clearly related with forest fuel. Be then more specific in line 217, "..vegetation communities changed towards more forest environemt"?

-If I am right, the melting of the glaciers mentioned in lines 220-221 are not well supported by the data – this would be shown by high detritic input into the lake during deglaciation, which is not the case (see 7e, highest forest, highest evaporation and lowest detritic input). Not clear. . .

-Lines 239-241. This is not clear – please rephrase.

-Lines 242-245. The vegetation shift towards more Pinus does not seem to be due higher continentality as stated here. Check Fig. 4, where the peak in Pinus seems to be reached during the lowest seasonal contrast (low summer insolation and high winter insolation – cooler summers and warmer winters).

I hope my comments help improving the manuscript.

---

## Editor Comment (EC1) · N. Combourieu Nebout (Editor) · 21 Feb 2017

Dear authors

We have now received the comment of three reviewers and an additional short comment . Both reviews are positive and underline the potential of your research and the interest of results. They give you clear indications about how the paper can be improved. You have to clarify some point and modify your manuscript taking into account very carefully the remarks and the proposed improvements.

Please post your replies to all the comments on the discussion forum that explain how

you want to modify your manuscript. Please also prepare a revised version of your paper accordingly. In the revised version, I would like to see your corrections in track change mode.

With my best wishes and looking forward to your responses and revised version.

Nathalie Combourieu-Nebout

---

## Author Comment (AC1) · 28 Mar 2017

Dear Nathalie Combourieu-Nebout,

Referring to your decision letter from 21st February 2017, I would like to resubmit on behalf my co-author Thomas Litt a revised version of our manuscript entitled 'A new-resolution pollen sequence at Lake Van, Turkey: Insights into penultimate interglacial-glacial climate change on vegetation history' to CP. We thank you and the reviewers for all constructive suggestions. We considered them all and implemented them in one revised 'track change mode' version and a second revised but 'unmarked' version.

[Figure]

Below please find details on how we address the reviewer comments. We think the revisions improved our manuscript significantly, and hope that our manuscript in its present form will be better suited for publication in CP.

Best regards, Nadine Pickarski

Please also note the supplement to this comment:
http://www.clim-past-discuss.net/cp-2016-133/cp-2016-133-AC1-supplement.zip

**Fig. 1.**

Fig. 2.

Fig. 3.

[Figure]

**Fig. 4.**

[Figure]

Paleovegetation — Oxygen isotopes

**Fig. 5.**

[Figure]

**Fig. 6.**

---

## Author Comment (AC2) · 28 Mar 2017

Dear Gonzalo Jiménez-Moreno,

Thank you very much for all helpful suggestions and useful recommendations to improve the quality of this manuscript. We considered all of them and implemented them in the revised manuscript.

Best regards, Nadine Pickarski

Please also note the supplement to this comment:

[Figure]

http://www.clim-past-discuss.net/cp-2016-133/cp-2016-133-AC2-supplement.zip

[Figure]

**Fig. 1.**

(a)

(b)

**Fig. 2.**

MIS

| 6 | 7a | 7c | 7e | 8 |

(a) LR04 (δ¹⁸O ‰ VPDB)

T II

T IIIA

T III

(b) Insolation 40°N (Wm⁻²)
Mid-June
Mid-January

(c) $\delta^{18}O_{bulk}$ (‰ VPDB)

T + δ¹⁸Oₓ

dry / high evaporation

(d) Ca/K ratio

low detrital input

(e) Paleofire intensity
Charcoal conc. (particles cm⁻³ x 100)

high fire intensity

(f) Paleovegetation
Total tree pollen (AP %)
dec. *Quercus*
*Pinus*

warm / humid conditions

PAZ IIIc6 IV1 IV2 IV3 IV4 IV5 IV6 Va1 Va2 Va3 Vb 1 Vc2 3 4 VI

130 135 140 145 150 155 160 165 170 175 180 185 190 195 200 205 210 215 220 225 230 235 240 245 250

Age (ka BP)

**Fig. 3.**

[Figure]

**Fig. 4.**

**Paleovegetation**

**Oxygen isotopes**

Fig. 5.

[Figure]

**Fig. 6.**

---

## Author Comment (AC3) · 28 Mar 2017

Dear anonymous referee,

Thank you very much for all helpful suggestions. We considered all of them and implemented them in the revised manuscript. We rewrote the manuscript more clearly and added further explanations of for how the Marine Isotope Stages were referred to terrestrial temperate interval in the Lake Van area. Additional description was also made for the chronology and the final climatostratigraphic alignment of the presented Lake Van sequence. We also paid more attention by using the term 'succession'. We

avoid the generalization of forest development for warm stages in the discussion.

Best regards, Nadine Pickarski

Please also note the supplement to this comment:
http://www.clim-past-discuss.net/cp-2016-133/cp-2016-133-AC3-supplement.zip

[Figure]

**Fig. 1.**

[Figure]

**Fig. 2.**

[Figure]

**Fig. 3.**

[Figure]

**Fig. 4.**

[Figure]

**Fig. 5.**

[Figure]

**Fig. 6.**

---

## Author Response (AR1)

**General comment:**
This is a very interesting work, clearly presented and well discussed. It provides new detailed information on the vegetation development and climate changes in the eastern Mediterranean region in the time interval 250-130 ka.

My main concern relates to the succession of vegetation types claimed for the warm stages in the paragraphs "forested periods" and "conclusions". At lines 212-214 it is stated that "the vegetation succession starts with the colonization of open habitats by pioneer trees, such as Betula, followed by sclerophyllous Pistacia cf. atlantica and a gradual expansion of deciduous Quercus". "The ensuing ecological succession at Lake Van is documented by high percentages of dry-tolerant and/or cold-adapted coniferous species (lines 235-236). This succession is not clearly visible in any of the forest phases: deciduous Quercus is always starting before or at the same time of Betula. Pistacia is almost missing in the forest phase corresponding to MIS 7 c. Pinus develops after Quercus in MIS 7e, but it is coeval in MIS 7c and is very sparse in MIS 7a. I suggest to avoid this generalization, which may be of interest in central Europe, far away from glacial refugia, but is not appropriate for the Lake Van region. Similarly, I do not find in the diagrams the most depleted (negative) delta18Obulk values at the base of each early temperate stage (lines 218-219): this is true at the onset of MIS 7e, but the next time interval with low values of delta18O is found during the cold stage of MIS 7d, and the following low values are recorded during the warm stage of MIS 7a. Thus, the suggested generalization is not convincing.

Dear Donatella Magri,

Thank you very much for your helpful suggestions and useful recommendations to improve the quality of this manuscript.

Concerning the succession of vegetation types in our manuscript, we avoid the generalization for warm stages in the paragraphs 'forested periods' and in the 'conclusions'. We revised the 'forested period' section and described the steppe-forest development for each terrestrial temperate interval. In addition, we also avoid the generalization of 'generally' low oxygen isotope values at the beginning of each temperate stage.

Best regards,
Nadine Pickarski

**Minor corrections:**

**Line 55:** Roucoux et al., 2008, 2011
Changed.

**Line 56:** Tzedakis et al., 2003b, 2006
Changed.

**Line 79:** there is no correspondence of rainfall values with table I. Besides, Van is not in the north-east (Erciˌs is in the NE)

Good remark, thank you! We checked the rainfall values and revised the sentence as follows: 'At Lake Van, rainfall decreases sharply from south-west (c. 1232 mm a$^{-1}$ in Bitlis) to north-east (c. 421 mm a$^{-1}$ in Erciş; Table 1)...' (now line 84-85).

**Line 82:** the vegetation cover around Lake Van
Done.

**Line 87:** dominated by dwarf-shrub steppes
Changed.

**Line 122:** HCl
Changed.

**Line 131:** diagram of selected taxa. You may consider to add here that the complete list of data will be available on PANGEA
We added the link to the PANGAEA database at the end of this section (now line 156-157).

**Line 139:** on bulk sediment samples
Done.

**Line 163:** (max. _70 %)
Thank you. It should be read: '…Chenopodiaceae (max. ~76), …'.

**Lines 212-213 and 235-237**: see general comments
We generally improved the section of 'forested periods' and avoid all generalization of the forest succession for the Lake Van region. Furthermore, we also avoided the phrases '… the most depleted oxygen isotope values occur at the start of each early temperate phase.' (Concerning the isotope signature see also to the detailed reply to Referee#3)
For better understanding, we added the Terminations (TIII at 250 ka, TIIIA at 222 ka, and TII at 136 ka) after Barker et al. (2011) and applied by Stockhecke et al. (2014) for the Lake Van sequence. Here, the beginning of TIIIA occurs right before the expansion of Poaceae, *Artemisia*, and the shift from positive to negative isotope values within PAZ Vb (MIS 7d).

**Line 236:** that suggest a cooling
Done.

**Line 240:** in other words: : :: : :this sentence seems incomplete
Due to general changes/improvements of this section ('forested periods'), we removed this rephrase.

**Line 250:** from a variety of factors
Removed. See reply above.

**Line 274:** species
Done.

**Line 280:** tolerates
Done.

**Line 347:** Between 193 and 157 ka BP
Done.

**Line 350:** an age of _189 ka BP is probably too precise with respect to the chronological setting of the record. Better say _189 ka BP
Changed.

**Line 371:** shares

Done.

**Line 373:** refers
We changed the word in 'refer'.

Line 408: colder but wetter climate conditions during MIS 6e than during MIS 3
Done.

**Line 422:** a unique record
Changed.

Lines 428-430: see general comments
See reply above.

**Line 439:** wetter conditions than during MIS 3
Done.

**Lines 441-444:** In the last sentence of the conclusions it would be better to emphasize what is new in your record instead of remarking what was already known
Thank you for the opportunity to comment on this manuscript. It concerns a sound data set of great value to the palaeoecological community as it comes from a region where such data for this period are scarce. Overall, I think the work is good and should be published in CPD but there are a number of important details that need to be considered and corrected first. In many instances these are related to terminology, definition of terms and ambiguity or circularity in the phrasing. One important example of this is the use of the term "steppe forest" without definition or explanation. Another is the use of marine isotope stage names to refer directly to intervals identified in the pollen record with no explanation for how that equivalence was established (even once an explanation is given, MIS terms should not be used directly for terrestrial intervals - see further details below). There is occasional circularity and a lack of clarity as to what was used to infer what but this is usually a question of the phrasing, and not a fundamental problem with the argument (examples below). I think that to make a convincing argument, the basis of both the stratigraphy and the chronology should be outlined in more detail (even if they are described elsewhere) so that the paper can stand alone. Without this, it is difficult to assess the validity of statements about the relative timing of events in the Lake Van pollen record and global scale climatic events. The vegetation reconstructions/inferences (particularly those involving trees) I think need to be more clearly described and the basis for the inferences better founded (e.g. with reference to modern pollen-vegetation studies, where possible). Also, I think there IS succession where the authors have suggested there is not... this could do with some more consideration. These issues are detailed below along with suggestions for grammatical corrections. In addition there are numerous minor grammatical errors (especially plural/singular, tenses) (not all are listed below).

Dear anonymous referee,

Thank you very much for your constructive suggestions. We considered all of them and implemented them in the revised manuscript. We rewrote the manuscript more clearly and improved the discussion section, especially, as recommended, the detailed vegetation inferences and successions.
Concerning the use of the term 'steppe-forest', we added the definition of an oak steppe-forest after Zohary (1973) and Frey and Kürschner (1989), which can also be described as a 'mixed formation of cold-deciduous broad-leaved montane woodland and xeromorphic dwarf-shrublands'.
In addition, we added further explanations in an extra paragraph of how the Marine Isotope Stages were referred to terrestrial temperate interval in the Lake Van area. Additional description was also made for the chronology and the final climatostratigraphic alignment of the presented Lake Van sequence.
We carefully checked our use of English. We overall think the revisions improved our manuscript significantly, and we hope that our manuscript in its present form will be better suited for publication.

Best regards,
Nadine Pickarski

**Page Line Comment**

**1 11** "effective moisture" needs some qualification (high, low?) otherwise the meaning is not clear
We added 'high' to classify the phrases 'effective moisture'.

**1 12** "forest" ought probably to be qualified with "open" since this is "steppe forest"

We rewrote the sentence as follows: 'Integration of all available proxies shows three intervals of high effective moisture availability, evidenced by the predominance of steppe-forested landscapes (oak steppe-forest),…' (now line 11-12).

**1 12** I think the conventional term for the biome is "wooded steppe" (e.g. Allen et al. 1999, Nature 400, 740 – 743). If "steppe forest" means something other than this, then it must be defined (and in any case, a reference is needed).

According to Zohary (1973), the southern mountain slopes are covered by the Kurdo-Zagrosian oak steppe-forest belt, containing *Quercus brantii*, *Q. ithaburensis*, *Q. libani*, *Q. robur*, *Q. petraea*, *Juniperus excelsa*, and *Pistacia atlantica*. This oak steppe-forest has also been described as 'mixed formation of cold-deciduous broad-leaved montane woodland and xeromorphic dwarf-shrublands' by Frey and Kürschner (1989). Furthermore, several pervious vegetation studies at Lake Van used the term 'oak steppe-forest', see also Zohary (1973); van Zeist and Bottema (1991); van Zeist and Woldring (1978); Wick et al. (2003). We added the definition of oak steppe-forest in the section 'Site description'.

**1 13** "The warmest stage: : :" The previous sentence suggests moisture is the main limiting factor. If temperature is important too, then need to make it clear that both are involved throughout the text (i.e. avoid summarising warmer/wetter as "warmer").

Changed in: The warmest/wettest stage as indicated by… (now line 15).

**1 13** "in terms of" I think this should read "as indicated by"

Changed.

**1 13** "amplitude" Double check – do you mean amplitude, or duration (or both)? Please clarify this.

In this context, I was referring to the amplitude of the penultimate interglacial, which was lower than during the next interstadial (MIS 7c).

**1 14** Insert ": : : the tree population maximum associated with: : :" before "MIS 7"

Done.

**1 17** Clarify presence or absence of trees in this instance of "steppe"

We replaced the term 'steppe landscape' by '…periods of treeless vegetation…'.

**1 19** Replace "more" with "higher"

Changed.

**1 21** The mild conditions inferred here are also in agreement with pollen records from elsewhere in southern Europe.

We added the agreement with other pollen records from southern Europe within this sentence. Now it reads: 'In contrast, the occurrence of higher temperate tree percentages throughout MIS 7b points to relatively mild conditions, which is in agreement with other pollen sequences in southern Europe.' (now line 22-25)

**1 25** Insert after "subdued oscillations": ": : : as in other records of this interval from southern Europe." E.g. MD01-2444 and I-284.

Done.

**1 27** Clarify what it is that indicates cooler and wetter conditions (it's not the identification of MIS 6e!)

We clarify the indication of cooler/wetter climate conditions. Now it reads: 'Furthermore, we are able to identify the MIS 6e event (c. 179-159 ka BP) as described in marine pollen records, which reveals clear climate variability due to rapid alternation in the vegetation cover.' (now line 32-34).

**2 36** Could you say what the resolution was in that study?

Done. Now it reads: 'Based on millennial-scale time resolution (between c. 1-4 ka), the 600,000 year old record already shows….' (now line 42).

**2 41** Replace "allow" with "have allowed"
Changed.

**2 45** Replace "is not being" with "has not been"
Changed.

**2 49** Replace "already available" with "existing"
Changed.

**2 57** Replace "this presented study" with "our"
Changed.

**2 57** Delete "want to"
Done.

**2 58** Change to past tense
Done.

**2 61** Change to past tense
Done.

**3 67** "meter" should be plural
Done.

**3 77** "latitudes" should be singular
Done.

**3 88** It would be helpful to know whether these forest and shrub formations represent the "natural" state of the vegetation versus the result of human impacts (e.g. pastoralism).
In line 82-83, now in line 88-89: We already mentioned in the sentence above, that the present-day vegetation cover around Lake Van was and is the results of agriculture and pastoralism.

**4 99** "those" is ambiguous: : : can you say what "those" refers to? (Existing pollen data?)
We rewrote the sentence as follows: 'In this section, we combine new pollen and isotope data with the already existing low-resolution pollen record published by Litt et al. (2014) and oxygen isotopes data derived from bulk sediments ($\delta^{18}O_{bulk}$) analyzed by Kwiecien et al. (2014).' (now line 105-107).

**4 101** Chronology section – perhaps the explanation of how (at least this part of) the Lake Van sequence has been aligned to the marine isotope stratigraphy belongs here?
Thank you very much for this advice. We added the following section: 'Marine isotope stage (MIS) boundaries follow Lisiecki and Raymo (2004). … For the climatostratigraphic alignment of the presented Lake Van sequence, the proxy records were visually synchronized to the speleothem-based synthetic Greenland record ($GL_{T-syn}$ from 116 to 400 ka BP; Barker et al., 2011) (now line 117-119).

**4 114** How were the age control points identified – in which proxy record?
We added further information about the 'age control points'. Now it reads: 'The identifications of TOC-rich sediments containing high Ca/K intensities and increased AP values at the onset of interstadials/interglacials were aligned to the interstadials/interglacial onsets of the synthetic Greenland record by using 'age control points'. Here, the correlation points of the Lake Van sedimentary record have been mainly defined by abiotic proxies (i.e., TOC) caused by a higher time resolution of this data set in comparison to the pollen samples available during that time.' (now line 119-124).

**5 127** Insert "group" after "taxonomic"
Done.

**5 131** "percentages" should be singular

Done.

**5 133** Should be "lake surface"?
Changed to '…to evaluate lake surface conditions, …'.

**5 137** Replace "was" with "were"
Done.

**5 142** Insert "were made" after "measurements"
Done.

**5 157** "deciduous forested": : : I think you need to specify whether this is closed or open forest because the implication of following this with "open steppic landscapes" is that the forest was closed canopy which, given the low AP%, is unlikely. Which leads to the next comment: : :
At Lake Van, the AP maxima do not exceed 50-60%, suggesting that 'closed' forest conditions were never established in eastern Anatolia. It is always an 'open' oak steppes-forest, similar to the potential present-day vegetation cover at the southern shore of Lake Van. It is a cold-deciduous broad-leaved montane woodland and xeromorphic dwarf-shrublands (Frey and Kürschner, 1989).
Here, we replace 'forest' with 'open deciduous oak steppe-forest. Now it reads: 'The pollen diagram provides a broad view of alternation between regional open deciduous oak steppe-forest and treeless desert-steppe vegetation.' (now line 177-178).

**5 159** With low AP % values, I'm not sure "forested" gives the right impression. It sounds a bit too, well, "forested"! Is there an alternative term that would be a better representation of the open landscape with few trees that the pollen data seem to represent? Ideally, this would have its basis in modern pollen-vegetation work.
See reply above. Now it reads: 'We were able to recognized three main phases (PAZ Va1, Va3, Vc2, and Vc3), where total arboreal pollen percentages reach above 30%.' (now line 178-179).
Regarding a modern pollen-vegetation work at Lake Van, at present there is only one monthly resolved pollen study available (Huguet et al., 2012), which was obtained from a sediment trap in the lake basin. However, this study does not reflect the mosaic-like vegetation at Lake Van.

**6 166-7** "The highest concentration peaks occur during forest intervals". Please rephrase this to remove the circularity. (How do we know these were forested intervals? Partly, because of the high pollen concentration!).
We revised this sentence in: 'During PAZ IV1-6, Va2, Vb, and VI, the pollen concentration is dominated mainly by steppic herbaceous pollen species (between 5,000 and 52,000 grains cm$^{-3}$), whereas PAZ IIIc 6, Va1, Va3, and Vc2-3 consist of tree and shrubs taxa (all above c. 5,000 grains cm$^{-3}$).' (now line 188-190).

**6 171** Add a brief comparison with the pollen record here (to be consistent with the next sentences about Pediastrum which are compared with the pollen).
To be consistent with the rest of this section, we revised the sentences. Now it reads: 'In total, six green algae taxa were identified in the Lake Van sediments. Fig. 2a presents only the most important *Pseudopediastrum* species. The density of the thermophilic taxa *Pseudopediastrum boryanum* reaches maxima values (c. 5,500 coenobia cm$^{-3}$) combined with high arboreal percentages especially during PAZ Vc2. In contrast, the cold-tolerant species *Pseudopediastrum kawraiskyi* occur during the treeless phases (PAZ IV4-2, max. values c. 2,000 coenobia cm$^{-3}$).' (now line 191-195).

**6 181** Is the amplitude exceptionally high? This phrasing suggests you have made comparisons with other records: : : if so, please indicate broadly which records (or kinds of records) it is high relative to.
We have not made any comparison with other records in this case. The Lake Van isotope composition shows a high-frequency oscillation. We replaced 'high amplitude' by 'high-frequency oscillation' (now line 207-208).

**7 197** Please say how you define stadial and interstadial here OR avoid using these terms here and make the correlation between particular peaks in the isotope curve and particular stadial-interstadial transitions (defined in other records) later on. I think the same applies for Termination III (since you haven't yet clearly justified the identification of TIII in the lake Van record).

Thank you very much for your good advice. We added a new section 'Boundary definition and biostratigraphy', where we defined Terminations, interglacials, interstadial/stadial stages, and the correlation to the Marine Isotope Stages (now line 227-248).

**7 204** This should read "marked" not "remarkable"

Done.

**7 205** Does "here" refer to this study? If so, the "generally considered" does not make sense. Please clarify.

We rephrase this section. Now it reads: 'According to Litt et al. (2014), the three-marked temperate arboreal pollen peaks (PAS Vc, Va) can be described as an interglacial complex. This general pattern of triplicate warm phases interrupted by two stadials (PAS Vb, PAZ Va2) is characteristic both in marine and ice-core records (MIS 7e, 7c, 7a after Lisiecki and Raymo, 2004), as well as for continental pollen sequences in southern Europe correlated and synchronized by Tzedakis et al. (2001).' (now line 250-254).

**7 206** The sentence starting "This general pattern: : :" is ungrammatical. The warm phases alternate with the cold phases. Please rephrase. Also, it is interesting that you do not mention changes in moisture availability here. There either needs to be a justification for that (deliberate?) omission, or both climatic parameters need to be considered.

Rephrased. See reply above.

**7 207** There is something odd about the line of reasoning here. On what basis did you establish the equivalence between the phases with more trees and MIS 7e, 7c and 7a if not by comparison of the pollen record with a marine isotope record (such as that used in the stratigraphy of Martinson et al. 1987) directly or indirectly? (I.e. "comparable with the marine classification by Martinson: : :" does not make sense). Also, take care with using the language of marine isotope stratigraphy to directly refer to intervals recorded by the pollen record – it is not strictly correct to do that (though of course we all do it informally). Ideally, wooded intervals in the AP% curve should not be directly aligned with the (apparently) equivalent MIS stages; there are significant offsets (and uncertainties) in the timing of the beginning and end of forest intervals on land relative to the beginning and end of warm marine isotope stages. Marine pollen records, form the Iberian margin and elsewhere, which combine marine isotope stratigraphy with a terrestrial vegetation signal are the only records in which the relative timing can be established directly (and these show significant offsets).

In agreement with Tzedakis (2007), the onset of terrestrial temperate intervals corresponds broadly with the Mid-June insolation maximum. Here, the length of the delay depending on local conditions keeping moisture availability below the tolerance threshold for tree growth.

However, we explained the basis how we aligned arboreal pollen record (as part of an independent proxy record) with the MIS stages in section 'Chronology' and 'Boundary definition and biostratigraphy' (See reply above). For the synchronization, we used the independent XRF and TOC proxy records that showed more or less no offsets in the timing of the beginning and end of warm Marine Isotope Stages. Therefore, we were able to combine the terrestrial vegetation signal, which documents (of course) offsets with the beginning and end of warm phases, with the MIS stratigraphy (See also reply **10 315**).

**7 214** Should "abrupt" be "brief"? In this context, "abrupt" doesn't really make sense.

Removed.

**7 217** Rephrase: link the sentence starting "It is clear: : :" to the previous one and remove "it", which is unclear.

Due to the general improvements of this section, the sentences is now revised as follows: '….fire activity rose at the beginning of each warm phase when global temperature increased and the vegetation communities changed from warm-productive grasslands to more steppe-forested environments.

Increased fire frequency is clear visible by high charcoal concentration up to 3,000 particles cm$^{-3}$…'
(now line 271-274).

**7 217** ": : : vegetation communities changed." State what kind/direction of changes this refers to.
See reply above.

**7 219** When discussing events in the past, not stratigraphy, the terminology is "start" not "base"
Changed.

**7 222-3** The inference of "oak steppe-forests where summer-green Quercus rises consistently above 20%" needs a few words of explanation and justification, and a reference.
We already define the term 'oak steppe-forest' in the section 'Site description'. See also reply to comment **1 12**.

**8 233** "this" should be "their" as in "their hypothesis", but it would also be helpful to have a brief re-statement of that hypothesis here.
For the better understanding, we added some additional information about the hypothesis from Kwiecien et al. (2014). Now it reads:
'Furthermore, Kwiecien et al. (2014) described the relation between soil erosion processes and the vegetation cover in the catchment area. They define interglacial conditions related to increased precipitation indicated by higher amount of arboreal pollen and lower detrital input. Our new high-resolution pollen record validates their hypothesis with high authigenic carbonate concentration (high Ca/K ratio; low terrestrial input) along with the increased terrestrial vegetation cover density (high AP percentages above 50%) during the climate optimum (c. 240-237 ka BP; Fig. 3).' (now line 289-294).

**8 241** I think there needs to be a clear statement of how these records were aligned – i.e. how do you know that the vegetation changes (that you interpret to represent cooling/drying) recorded in the Lake Van sequences occurred BEFORE ": : : ice accumulation is evident: : : in MD01: : : "?
We removed this section.

**8 241** The linking phrase ("In light of these insights: : :") does not work because the insights just described are not what suggests a shift from temperate to coniferous taxa.
We removed this phrase due to general modifications of the text.

**8 248** Why "re-expansion" not just "expansion" (implies a second expansion)?
Changed.

**8 253** The persistence of relatively large tree populations through the period equivalent to MIS 7b was noted at Lac du Bouchet and at Ioannina; please cite this work here.
Done.

**8 263** MIS 7c is not an interstadial: : : unless you want to define it as such at Lake Van (but then this must be explained and justified).
We improved the section 'forested periods' and added some more information/comments about the penultimate interglacial complex MIS7, including MIS 7c and MIS 7a as an interglacial stage.

**9 263-266** All good reasons listed here for not calling MIS 7c an interstadial.
You are completely right. It was a misunderstanding. MIS 7e, 7c, and 7a are, of course, interglacial stages.

**9 273** Which other tree taxa are missing, besides Pistacia, from the succession: : : I couldn't see any others. If only Pistacia is missing from the wooded interval equivalent to MIS 7c, this is not sufficient to say there is no succession. I think there is: as in the "7e" interval the "7c" tree population expansions begin with Betula, continue with Quercus and this is followed by expansion of Pinus populations.
Good remark! We rewrite the complete section 'forested periods'. See reply above.

**9 275** Ensure the phrasing reflects the fact that you are describing conditions in the region of lake Van and that the same conclusions may not apply elsewhere (i.e. include reference to the region to which your conclusion applies).
We removed this sentence.

**9 277** Don't need BP with ka, conventionally.
Done.

**9 277** Along with the intervals that have more trees, the open (treeless) intervals also need to have their equivalence to the marine isotope stratigraphy justified. To repeat – it is not good practice to refer directly to intervals identified in a terrestrial pollen record with the MIS nomenclature (you need to demonstrate the basis for the correlation, and even then, I would still say "the interval broadly equivalent to MIS: : :" or similar wording).
See also reply line **7 207.**

**9 277** Related to the comment above, replace "MIS 7d" with "pollen record between : : : and : : : ka" or use zone names. There are numerous other places in the manuscript where MIS terminology is used where it is not appropriate.
We replace MIS 7d with: 'The two periods between the three temperate forested intervals, PAZ Vb (227-221 ka, 109.1-106.5 mcblf) and PAS Va2 (208-203 ka, 101.3-99.9 mcblf), are broadly equivalent to MIS 7d and MIS 7a.' (now line 367-369).

**9 290** Please give references (after ": : : Lebanon and southern Europe.")
Done.

**9 293-4** This description of the vegetation during the interval equivalent to MIS 7b is not consistent with the description of this interval above (where 7d and 7b, to use the informal shorthand, are described together as having "extensive steppe vegetation: : : [and] inhibited tree growth: : :"
Thank you very much for this comment. We improved the description of the vegetation during cold periods and paid attention on exact wording. Now it reads: 'At Lake Van, cold periods are generally characterized by: (I) extensive steppe vegetation when tree growth was inhibited either by dry/cold or low atmospheric $CO_2$ conditions (Litt et al., 2014; Pickarski et al., 2015b), ….' and 'In contrast to conventional cold periods at Lake Van, the second phase (PAS Va2) recognizes only a slight and short-term steppe-forest contraction.'.

**9 297** Why is higher in ': : :'? (another occurrence in line 304: 'high')
We deleted both '….'.

**10 299** Delete "arboreal"
Done.

**10 300** "i.e." should be "e.g." here
Done.

**10 305** Check – if CO2 was higher in 7b, it is more likely to have been warmer than 7d.
You are right. The $CO_2$ content during MIS 7b was a bit higher (c. 230-240 ppm) than during MIS 7d (c. 207-215 ppm).

**10 308** and onwards Consider using past tense in this section as it discusses events in the past rather than the record of those events.
We have paid more attention to the use of correct tenses.

**10 315** Delay relative to what: : :?
…relative to the glacial/interglacial boundary as defined in NGRIP and $GL_{T-syn}$. We revised this sentence. Now it reads:

'….the MIS 8/7e, MIS 7d/7c as well as the MIS 6/5e boundary in the continental, semi-arid Lake Van region recognized a delayed expansion of deciduous oak steppe-forest of c. 5,000 to 2,000 years, comparable to the pollen investigations of the marine sediment cores west of Portugal by Sánchez Goñi et al. (2002, 1999). As already shown in high-resolution Lake Van pollen studies by Wick et al. (2003), Litt et al. (2009), and Pickarski et al. (2015a), a delay in temperate oak steppe-forest refer to the Pleistocene/Holocene boundary as defined in the Greenland ice core from NorthGRIP stratotype (for the Pleistocene/Holocene boundary; Walker et al., 2009) as well as from the speleothem-based synthetic Greenland record (GL$_{T-syn}$; Barker et al., 2011; Stockhecke et al., 2014) can be recognized. (now line 409-417) (see also reply to comment **7 207**).

**10 318** Replace "due" with "indicated by"
Done.

**10 324** "However: : :" doesn't make sense here.
We rephrase this sentence as follows: 'Compared to *Carpinus betulus*, deciduous oaks are….'.

**10 327** Reference required (to support observation about range of ecological requirements within the Quercus genus).
In general, we have added some additional information about the ecological requirements of dec. *Quercus* and added relevant references (see also reply below)

**10 328** There seem to be some logical steps missing: : : can this be explained more clearly? Make clear that both abundance and composition of tree populations differs. Also, it is necessary to reconcile this argument for wetter/cooler conditions with the presence of Pistacia close to the start of the "forest" interval corresponding to MIS 7e.
See also reply above. We added further information to close the missing steps.
Now it reads: '…*Carpinus betulus* usually requires high amounts of annual rainfall (high atmospheric humidity), relatively high annual summer temperature, and is intolerance of late frost (Desprat et al., 2006; Huntley and Birks, 1983). In oak-hornbeam communities, *Carpinus betulus* is replaced as the soils are relatively dry and warm or too wet (Eaton et al., 2016). Compared to the common hornbeam, deciduous *Quercus* species are 'less' sensitive to summer droughts (even below 600 mm/a; Tzedakis, 2007), and therefore, a decrease in soil moisture availability would favor the development of deciduous oaks (Huntley and Birks, 1983). Especially, the deep penetrating roots of *Quercus petraea* allow them to withstand moderate droughts by accessing deeper water (Eaton et al., 2016). However, a variation in temperature is difficult to assess because deciduous oaks at Lake Van include many species (e.g., *Quercus brantii, Q. ithaburensis, Q. libani, Q. robur, Q. petraea*) with different ecological requirements (e.g., San-Miguel-Ayanz et al., 2016). Finally, the absence of *Carpinus betulus*, the overall smaller abundances of temperate trees (e.g., *Ulmus*), and the general low diversity within the temperate tree populations during the climate optimum of the first penultimate interglacial compared to the last interglacial indicates warm but drier climate conditions (similar to the Holocene).' (now line 429-444).

**11 335-6** This assumes that the "climate optimum" is equivalent to the "terrestrial temperate interval" – either justify this equivalents or use "terrestrial temperate interval" both times.
In this case, we use the term 'terrestrial temperate interval' for both times.

**11 341** Replace "evident" with "suggested" or "indicated"
Done.

**11 343** The "rapid decline in temperate trees" does not make sense: : : which decline does this refer to?
Revised to: 'Such observed climate deterioration is suggested by the dominance of semi-desert plants (e.g., *Artemisia*, Chenopodiaceae) and by the decline in temperate trees (mainly deciduous *Quercus* <5%) similar to that of the last glacial at the same site.' (now line 459-461).

**11 350-351** ": : : resembles the pattern of interstadial to stadial stages." - as defined by what?

The interstadial/stadial pattern (e.g., Dansgaard-Oeschger events) was defined in the Greenland ice core record, esp. by, e.g., NGRIP, 2004; Rasmussen et al., 2014 for the last glacial period. We added some references and revised the sentence.

**11 359** A landscape cannot be "less extensive"
We removed this phrase.

**11 360** "greater" would make more sense that "great" here.
Done.

**11 362** Replace "values" with "populations" as this is about inferred vegetation now, rather than the pollen record.
Done.

**12 370** Should also cite Margari et al. 2010 for the Iberian margin marine pollen record of MIS 6.
Done.

**12 379** : : : and this pattern is also recorded at Lake Van?
What we see is that the Ca/K ratio (and also the TOC record at Lake Van, which is not mentioned in the manuscript) documents slight change to lower erosion processes around 150 ka (We have added this observation to the manuscript). I think the vegetation signal is to weak/subdued in an overall cold/dry climate to see any small changes in the record.

**12 385** Which transition does this refer to? Or should it read "transitions"?
Changed.

**12 394** At face value, this should not be the only reference given for the DO events 17 to 12.
We added further reference for Dansgaard-Oeschger events.

**12 396** Not clear what is meant by "compared to": : : do you mean "comparable with" or "similar to"? Or something else?
We replaced 'compared to' with 'comparable with'.

**12 397** "Intensities" is ambiguous: : : should it be amplitude?
Changed.

**13 403** "is supported by" should be "suggests" if the pollen forms the basis of this climatic inference.
Changed.

**13 409** ": : : points to a general picture of cold but 'wet' conditions during MIS 6e than experienced during MIS 3." This is not grammatically correct.
We revised this sentence as follows: 'Nevertheless, the occurrence of *Pinus*, *Ephedra distachya*-type as well as the cold-tolerant algae *Pseudopediastrum kawraiskyi* indicates colder/wetter climate conditions during MIS 6e compared to MIS 3.' (now line 527-529).

**13 426-7** It is not clear to me what this vegetation formation would look like. Which aspect was "dense"? I think the term "steppe" is incompatible with the term "dense forest" unless the two kinds of vegetation occurred simultaneously in different areas (e.g. open steppes with discrete areas of dense forest: : : but that wouldn't be called a "steppe forest")
We replaced 'dense' by 'well-developed'. See also reply Referee #2.

**13-14 431-2** ": : :strong thermal and hydrological seasonal contrasts during the last interglacial, and a higher humidity during the Holocene climate optimum: : :" are not discussed in the rest of the manuscript. If they are to appear in the conclusions, they need to appear earlier in the text as well.
You are right. This topic was not discussed in the manuscript. Therefore, we removed this sentence.

**15 453** Check spelling of Miriam: : :

Thank you very much for marking the typing error.

**20 Fig 2** Please add an indication of the basis on which the taxa shown were selected (ecological importance, abundant: : : ?). It would be helpful to know how many AP taxa are not included (and what proportion of the sum this represents). A curve for "other AP" would demonstrate this (if they are too rare to show, then this needs to be said). The same point applies to the NAP. Also, the curves are black (the fill is white).

We summed all other arboreal taxa in an 'Other AP' curve. We did the same for all other non-arboreal taxa 'Other NAP'. For a better distinction, all arboreal pollen curves are marked in black, whereas all non-arboreal curves are presented in grey. All exaggerations are marked in white (fill).

**21 Fig 3** Please indicate on what basis the MIS equivalents are assigned. Even if this is addressed in another paper, for this paper to make a convincing case, it needs to be said here too.

We revised this figure. We added the LR04 isotopic record after Lisiecke and Raymo (2004) and rewrote the caption as follows: 'Comparative study of Lake Van paleoenvironmental proxies during the penultimate interglacial-glacial cycle. (a) LR04 isotopic record (in ‰ VPDB) with Marine Isotope Stage (MIS) boundaries (grey bars) following Lisiecki and Raymo (2004); (b) Insolation values (40°N, Wm$^{-2}$) after Berger (1978) and Berger et al. (2007); (c) Lake Van oxygen isotope records $\delta^{18}O_{bulk}$ (‰ VPDB; new analyzed isotope data including the already published isotope record by Kwiecien et al., 2014); (d) Calcium/potassium ratio (Ca/K) after Kwiecien et al. (2014); (e) Fire intensity at Lake Van (>20 µm, charcoal concentration in particles cm$^{-3}$); (f) Selected tree percentages (total arboreal pollen (AP), deciduous *Quercus*, and *Pinus*) including the pollen data from Litt et al. (2014); PAZ – Pollen assemblage zone. Termination III at 250 ka, TIIIA at 223 ka and TII at 136 ka are indicated after Barker et al. (2011) and Stockhecke et al. (2014a).'

**22 Fig 4** Add a statement to explain on what basis the interglacials illustrated here (MIS 5e, 7e) are defined (because under some definitions, 7c could also be an interglacial).

Due to general improvements of the chapter '4.2 The penultimate interglacial complex', we added the definitions of interglacial/interstadials as well as the correlation with terrestrial temperate intervals with Marine Isotope Stages in the discussion (see replies above).

**23 Fig 5** Inclusion of AP-PJB% from Ioannina (as well as AP%) would have been more informative as this signal is more sensitive to climatic fluctuation and picks out a very similar pattern to that in lake Van: : : e.g. the minor decline of temperate tree populations associated with MIS 7b and a post-MIS 7a millennial scale oscillation.

Caption: Is it a "correlation scheme" if each curve is presented on its own timescale? There are some pronounced offsets in the timing of major vegetation changes which seem too large to be real and are likely to be exaggerated by age uncertainties.

Could you clarify how this diagram was constructed (in the caption if not in the text), where timescales align NECESSARILY (because of the way the age models have been developed, for example) and where timescales are the original published ones (and the sources for those age models: : : for example, have you placed the I-284 curve on the GL synth timescale, or on the timescale published in 2008?). Without this kind of information, it is difficult for the reader to understand the significance of apparent alignments and offsets.

We have added the AP-PJB% pollen curve from Ioannina, because this curve shows some new information to climate changes/fluctuation during the penultimate glacial. We also added the AP-PJ% pollen curve from Tenaghi Philippon to the figure to get additional information about regional climate fluctuations.

Concerning the different timescales of each climate archive, we revised the caption as follows: 'Comparison of Lake Van pollen archive with terrestrial, marine and ice core paleoclimatic sequences on their own timescales.'

This is an exciting and valuable new data set and a major contribution to the knowledge of past climate and vegetation development in the Middle East.

As you clearly show in your discussion, vegetational development, i.e. transitions from open steppe vegetation to various stages of deciduous and coniferous woodland (and vice versa) are not only driven by temperature, but largely by moisture sources and availability. Thus it is somewhat risky to directly relate the Lake Van pollen and oxygen isotope records to the marine isotope stratigraphy and to use the MIS terminology. I suggest to interpret the Lake Van record with regard to regional climate and vegetational change and use it as a basis for discussing possible correlations to the MIS, insolation etc. Be careful with the term 'succession'; I think it may not be used in a central European sense. At Lake Van there is a distinct gradient in moisture from south-west to northeast; the 'succession' from open steppe to deciduous oak woodland as described here might rather be a movement of the different vegetation formations from SW to NE than an all-over woodland expansion.

Dear anonymous referee,

Thank you very much for all helpful suggestions. We considered all of them and implemented them in the revised manuscript. We rewrote the manuscript more clearly and added further explanations of for how the Marine Isotope Stages were referred to terrestrial temperate interval in the Lake Van area. Additional description was also made for the chronology and the final climatostratigraphic alignment of the presented Lake Van sequence. We also paid more attention by using the term 'succession'. We avoid the generalization of forest development for warm stages in the discussion.

Best regards,
Nadine Pickarski

**Minor improvements and suggestions:**

(Linguistic and grammatical improvements suggested by referee#1 are not repeted here)

**Line 114:** please provide some additional information on how you did the synchronization
We revised the 'Chronology' section as follows (see also response to Referee #1; comment **7 101** and **7 114**):
'For the climatostratigraphic alignment of the presented Lake Van sequence, the proxy records were visually synchronized to the speleothem-based synthetic Greenland record ($GL_{T-syn}$ from 116 to 400 ka BP; Barker et al., 2011). The identifications of TOC-rich sediments containing high Ca/K intensities and increased AP values at the onset of interstadials/interglacials were aligned to the interstadials/interglacial onsets of the Greenland record by using 'age control points'. Here, the correlation points of the Lake Van sedimentary record have been mainly defined by abiotic proxies (i.e., TOC) caused by a higher time resolution of this data set in comparison to the pollen samples available during that time.' (now line 117-124).

**Lines 157 ff.:** you may replace 'forest' by 'woodland', '(sparsely) wooded landscape' ....

We rephrased the sentence (see also comment Referee#1). Now it reads: 'The pollen diagram provides a broad view of alternation between regional open deciduous oak steppe-forest and treeless desert-steppe vegetation.' (now line 177-178).

**Line 163:** Chenopodiaceae max. 70%
See also comment of Donatella Magri. It should be read: …(max. ~76)…

**Line 217:** how did the vegetation change?
We added some additional information about the vegetation change (see also Referee#1). Now it reads: 'Furthermore, the fire activity rose at the beginning of each warm phase when global temperature increased and the vegetation communities changed from warm-productive grasslands to more steppe-forested environments.' (now line 271-273).

**Lines 231-234:** please give a brief description of this relationship
We revised this section and added some more information about the relationship between erosion and vegetation cover. Now it reads:
'Furthermore, Kwiecien et al. (2014) described the relation between soil erosion processes and the vegetation cover in the catchment area. They define interglacial conditions related to increased precipitation indicated by higher amount of arboreal pollen and lower detrital input. Our new high-resolution pollen record validates their hypothesis with high authigenic carbonate concentration (high Ca/K ratio; low terrestrial input) along with the increased terrestrial vegetation cover density (high AP percentages above 50%) during the climate optimum (c. 240-237 ka BP).' (now line 289-294).

**Lines 239-241:** What is the link from the Lake Van vegetation to the MD01-2447 record based on?
Removed.

**Line 315:** delayed - relative to what?
…relative to the glacial/interglacial boundary as defined in NGRIP and GL$_{T-syn}$. We revised this sentence as follows:
'….the MIS 8/7e, MIS 7d/7c as well as the MIS 6/5e boundary in the continental, semi-arid Lake Van region recognized a delayed expansion of deciduous oak steppe-forest of c. 5,000 to 2,000 years, comparable to the pollen investigations of the marine sediment cores west of Portugal by Sánchez Goñi et al. (2002, 1999). As already shown in high-resolution Lake Van pollen studies by Wick et al. (2003), Litt et al. (2009), and Pickarski et al. (2015a), a delay in temperate oak steppe-forest refer to the Pleistocene/Holocene boundary as defined in the Greenland ice core from NorthGRIP stratotype (for the Pleistocene/Holocene boundary; Walker et al., 2009) as well as from the speleothem-based synthetic Greenland record (GL$_{T-syn}$; Barker et al., 2011; Stockhecke et al., 2014) can be recognized.' (see also reply Referee#1).

**Line 347:** replace 'during' by 'between'
Changed.

**Line 379:** Do you have any idea, why this evidence is missing at Lake Van?
If you wanted to know why other archives (e.g., Tenaghi Philippon) can recognized another period of abrupt warming between 155 and 150 ka and the Lake Van pollen record not, I can't give you a satisfactory answer. What we see is that the Ca/K ratio (and also the TOC record of Lake Van, which is not mentioned in the manuscript) documents slight change to lower erosion processes around 150 ka (We have added this fact to the manuscript). I think the vegetation signal is to weak/subdued in an overall cold/dry climate to see any small changes in the record. (see also reply Referee#1).

**Line 426:** 'dense' does not really fit with a steppe-forest - maybe 'well developed'
Changed.

**Lines 441-444:** Please add afew words saying what is different / new / special at Lake Van

We rewrote the conclusions and added what is new in our Lake Van record. See also reply to the comment of Donatella Magri.

**Fig. 2b:** Why is Thalictrum in the aquatic group? there are about 30 species in Antolia, most of them adapted to dry conditions, some prefer humid soils, but there is no real aquatic species.
You are completely right. We have grouped the species *Thalictrum* to the herbs.
This is an interesting article that shows new detailed pollen and oxygen isotope data from the MIS 8-6 part of the Lake Van sedimentary record. The authors interpret the pollen and isotope changes as changes in vegetation and precipitation/evapotranspiration around the lake basin. Vegetation changes between forested-steppe environments can be correlated with climate oscillations (interglacial/interstadials-glacial/stadials) described in the marine isotope records. The paper is well-written and the data support the interpretations/conclusions and thus deserves publication in CP.

However, in my opinion, there are some changes that need to be done before publication and several topics are not very well discussed in the manuscript and need to be clarified.

Dear Gonzalo Jiménez-Moreno,

Thank you very much for all helpful suggestions and useful recommendations to improve the quality of this manuscript. We considered all of them and implemented them in the revised manuscript.

Best regards,
Nadine Pickarski

**Below are my comments:**

It is not very clear what is really triggering the vegetation changes in the area – is it mostly temperature or precipitation? In some parts of the text temperature is indicated as the main trigger and in some others is precipitation or effective precipitation (supported by the isotope data). A clear example is the Abstract (lines 11-13) where effective precipitation is first introduced as the main trigger and then temperature...and this is very confusing as maximum insolation and thus maximum temperature would reduce the effective precipitation and should not produce the same effect on the vegetation. For example, in line 13 – maximum forest development during stage 7c does not seem to occur during summer insolation maxima…

First of all, each terrestrial temperate interval at Lake Van begins at the time of maximum summer insolation, which is the case during the penultimate interglacial (except for the youngest warm period, MIS 7a), last interglacial, and the current interglacial (see Fig. 4). This can be seen in the changes of abiotic proxies. The vegetation, however, reacts very slowly to climatic changes. The time lag of oak steppe-forest expansion depends mostly on spring/summer-drought conditions and/or by slow migration rates form refugia (This topic is now discussed in the section 'Comparison of past interglacials at Lake Van').

However, the most important trigger for vegetation changes in this semi-arid region is precipitation rates, esp. at the beginning of each terrestrial temperate interval. However, we also have to keep in mind that temperature changes have 'some' influences on the vegetation (in general).

Now, the text at this place is rephrased as follows: 'Integration of all available proxies shows three temperate intervals of high effective soil moisture availability, evidenced by the predominance of open forested landscapes (oak steppe-forest) similar to the present interglacial vegetation in this sensitive semi-arid region between the Black Sea, Caspian Sea, and Mediterranean Sea.

The wettest/warmest stage as indicated by highest temperate tree percentages can be broadly correlated with MIS 7c, while the amplitude of tree population maximum during the penultimate interglacial (MIS 7e) appears to be reduced due to warm but drier climate conditions.' (now line 11-17).

In this area, where precipitation is not very abundant I would think that forest development would be mostly related to precipitation or effective precipitation. I think you should be consistent throughout the text.

You are completely right. To be consistent throughout the manuscript, we paid attention to these phrases.

I also had the feeling that after reading the text and looking at the figures one still lacks of a clear idea of what is the relationship between insolation and plant dynamics in this record. In lines 268-269 it is stated that "…vegetation development (forest?) is clearly controlled by insolation forcing and associated climate regimes (high summer temperature, high winter precipitation)". I understand here that forest development in this area is "clearly" controlled by summer insolation, so in a very simplistic model if we had high summer insolation we would have had high forest development. This is a model that can be applied to several long Mediterranean records (see Tzedakis et al., 2007). However, if we look at figure 3, the major forest development seems to happen during summer insolation minima, so completely the opposite of what it is said in the text. Check stages 7e and 7c. What I understand from this is that forest cannot develop during periods of insolation maxima (and probably precipitation maxima) due to very high evaporation and that would explain the big lag between them. The vaguely mentioned lag in the text (line 315) is not just 2-3 ka…but about 10 ka (ie. stage 7c). This subject should be further explained and clarified in the text.

The maximum oak steppe-forest development occurred during summer insolation minimum, however, the start of forest development is closely associated with the timing of summer insolation peak.

Concerning the time lag between the start of interglacial conditions and the expansion of temperate trees, we have added some additional information (see also comments above, and to Referee#1, line **7 207** and **10 315** and Referee#2). We revised this section as follows:

'…the MIS 8/7e, MIS 7d/7c as well as the MIS 6/5e boundary in the continental, semi-arid Lake Van region recognized a delayed expansion of deciduous oak steppe-forest of c. 5,000 to 2,000 years, comparable to the pollen investigations of the marine sediment cores west of Portugal by Sánchez Goñi et al. (2002, 1999). As already shown in high-resolution Lake Van pollen studies by Wick et al. (2003), Litt et al. (2009), and Pickarski et al. (2015a), a delay in temperate oak steppe-forest refer to the Pleistocene/Holocene boundary as defined in the Greenland ice core from NorthGRIP stratotype (for the Pleistocene/Holocene boundary; Walker et al., 2009) as well as from the speleothem-based synthetic Greenland record ($GL_{T-syn}$; Barker et al., 2011; Stockhecke et al., 2014) can be recognized. The time lag of oak steppe-forest can be explained by slow migration of deciduous trees from arboreal refugia (probably the Caucasus region) and/or by changes in seasonality of effective precipitation rates (Pickarski et al., 2015a). In particular oak species are strongly dependent on spring precipitation (El-Moslimany, 1986). A reduction of spring rainfall and extension of summer-dry conditions favoured the rapid development of a grass-dominated landscape (mainly *Artemisia*, Poaceae; Fig. 2b) and *Pistacia* shrubs in the very sparsely wooded slopes (Asouti and Kabukcu, 2014; Djamali et al., 2010). Furthermore, high intensity of wildfires of late-summer grasslands, at the beginning of each warm period could be responsible for a delayed re-advance of steppe-forest in eastern Anatolia (Pickarski et al., 2015a; Turner et al., 2010; Wick et al., 2003).'

I am also puzzled about the isotope record from the lake and the comparison with the pollen data. First, if the interpretation of the data is correct (higher values, higher evaporation/dryness), the isotope data do not seem to agree with the summer insolation and it should. Second, if the vegetation was delayed because during summer insolation maxima there was too much evaporation, this would show a delay between the isotope data and the pollen and they basically covariate (except for some periods (stage 7a). Please clarify.

First, you are right. High oxygen isotope values indicate higher evaporation and/or dryness in the Lake Van area. In general, the interpretation of lacustrine stable isotope values at Lake Van is not as simple as in the marine record. It was analyzed from lacustrine bulk sediments, where all complex relationships, which are involved in the lacustrine carbonate precipitation, are not fully understood yet.

The isotope signature reflects several regional climatic variables as well as local factors, such as precipitation (rainfall, snow) and evaporation processes. They were also influenced by the water temperature and composition of the lake water. Therefore, the interpretation of stable isotope data at Lake Van is not that easy. Previous studies at Lake Van by Litt et al. (2009) and Wick et al. (2003) have found out that the depleted diluted isotope values at the beginning of terrestrial temperate intervals, esp. at the beginning of the Holocene, mainly reflects freshwater input due to snowmelt from local glaciers in the catchment area. This mechanism was transferred to earlier periods/interglacial onsets by Kwiecien et al. (2014). Unfortunately, at some points, this mechanism does not match (in particular in MIS 7a). Second, the delay of vegetation depends on local conditions keeping moisture availability below the tolerance threshold for tree growth in the more ecologically stressed areas. In the eastern Mediterranean area, the precipitation is still concentrated in the winter months, while the expansion of deciduous oaks is often hindered due to spring/summer-drought conditions at the beginning of interglacials (see reply above).

The fact that stage 7c shows one of the largest forest development in the record needs to be highlighted in the chapter about "Comparison of past interglacials at Lake Van".
We highlighted that the MIS 7c documents the largest oak steppe-forest development within the penultimate interglacial complex.

It is very confusing to see terms such as "steppe forest landscape" "oak-pine steppe forest" or "oak steppe forest" as these two terms "forest" and "steppe" are quite opposite. Why not calling these forests with AP pollen percentages around 60% "forests" or if you do not agree that they are close forests "open forests"? Also, steppes are mostly characterized in the area by Artemisia and Amaranthaceae, and Poaceae seems to be relatively abundant during the "forest" periods so it would not be quite an "steppe" environment.
According to Zohary (1973), the southern mountain slopes are covered by the Kurdo-Zagrosian oak steppe-forest belt, containing several oak species, *Juniperus excelsa*, and *Pistacia atlantica*. This oak steppe-forest has also been described as 'mixed formation of cold-deciduous broad-leaved montane woodland and xeromorphic dwarf-shrublands' by Frey and Kürschner (1989). Furthermore, several pervious vegetation studies at Lake Van used the term 'oak steppe-forest', see also Zohary (1973); van Zeist and Bottema (1991); van Zeist and Woldring (1978); Wick et al. (2003). We added the definition of oak steppe-forest in the section 'regional setting'. (See also reply Referee #1)

Even though there is certain variability during MIS 6 the forest oscillations are only between 0-10%. I would not call these oscillations "pronounced" as stated in the abstract (line 23). The authors should soften the language regarding these oscillations (section 4.2).
Here, we wanted to say that the early stage (c.193-157 ka BP) oscillates a bit more than the later stage (c. 157-131 ka BP). However, it was probably a bit exaggerated. We replace the phrase 'pronounced oscillations' by 'higher oscillations'.
We also softened the language regarding the 'pronounced' oscillations in section '4.3 The penultimate glacial' as well as in the 'Abstract' and in the 'Conclusion'.

Line 10: "The presented record displays the highest temporal resolution for this interval"? from where? Lake Van? Turkey? The World? Please be specific.
We added 'Lake Van'. Now it reads: 'The presented Lake Van pollen record displays the highest temporal resolution….'.

Pinus has an important role in the observed vegetation changes in this record and probably were important tree taxa regionally as well. Therefore, I think the authors should give some information about Pinus distribution in the area or regionally at Present in "Site description" as later it is mentioned that was transported to the area by the wind (lines 235-238).
Today, the distribution of *Pinus* (probably *P. nigra*) is located in the more continental western and central Anatolia areas. In eastern Anatolia and in the vicinity of Lake Van, *Pinus* is almost absent in the vegetation composition. Therefore, we do not give any further information about the *Pinus* distribution in the section 'Site description' to avoid any confusion.
However, we have added some more information in the discussion section. Now it reads: 'The ensuing ecological succession of the first warm stage is documented by a shift from deciduous oak steppe-forest towards the predominance of dry-tolerant and/or cold-adapted conifer taxa (e.g., *Pinus* and *Juniperus*; c. 237-231 ka). Especially, high percentages of *Pinus* suggest a cooling/drying trend, which occurred during low seasonal contrasts (low summer insolation and high winter insolation; Fig. 3). *Pinus* (probably *Pinus nigra*) as a main arboreal component of the 'Xero-Euxinian steppe-forest' recently occurs in more continental western and central Anatolia, and in the rain shadow of the coastal Pontic mountain range (van Zeist and Bottema, 1991; Zohary, 1973). Compared to the present distribution of *Pinus nigra* in Anatolia, the Lake Van region was probably more affected by an extended distribution area of pine during the penultimate interglacial as indicated by higher pollen percentages (Holocene below 5%; PAZ Vc2 up to 26%; PAZ Va3 up to 20%; Fig. 4). Holocene pine pollen was mainly transported over several kilometers via wind into the Lake Van basin.' (now line 295-305).

Line 120: Give unit for the "4 cm" size samples - cm3 ?
It was already written '…samples of 4 cm³….'. (now line 136)

I think the presence of Spiniferites should be better explained as many people would interpret this taxa as marine species. Do they occur in lake environments? Under what circumstances?
We added some further environmental information about the presence of *Spiniferites* spp.. Now it reads: 'Furthermore, we calculated dinoflagellate concentration (probably *Spiniferites bentorii*; cysts cm-3) in order to get additional information about environmental conditions of the lake water (Dale, 2001; Shumilovskikh et al., 2012; Fig. 2a). The occurrence of *Spiniferites* spp. in lacustrine sediments suggests low aquatic bio-productivity (low nutrient level) and hypersaline conditions (Zonneveld and Pospelova, 2015; Zonneveld et al., 2013). In this study, the concentration of dinoflagellate cysts is high (500-2,000 cysts cm-3) during non-forested periods, especially within PAZ IV1, IV3, IV5, Va2, and PAS Vb.' (now line 196-201)

Lines 195-196: "The d18O composition of the lake water becomes progressively more enriched during interglacial/interstadial periods". Not fully true – check stage 7a where the opposite happened. Please be more specific.
We rephrase this sentence as follow: 'At the beginning of major forested phases (e.g., PAZ Vc4, the end of Vb, Va1, and IIIc6), the $\delta^{18}O_{bulk}$ composition of the lake water becomes more depleted (Fig. 3c). According to Kwiecien et al. (2014) and Roberts et al. (2008), negative isotope values document not only enhanced precipitation during winter months but also the significant contribution of depleted (diluted) snow melt/glacier meltwater during the summer months.' (now line 221-225) (see also reply above).

**Lines 197-200:** 'Termination III (T III at 241.4 ka BP) and at the transition from stadial to pronounced interstadial periods documents not only enhanced precipitation during winter months but also the significant contribution of depleted snow melt/glacier meltwater during the summer months (Kwiecien et al., 2014; Roberts et al., 2008).' – This statement is not clear – in Fig. 3 the isotopic changes are explained and changes in dryness or evapotranspiration, supported by low detritic input in the lake.
See reply above.
Here, enhanced freshwater input and/or precipitation is supported by high detrital input (see also reply below).

The charcoal record is clearly related with forest fuel. Be then more specific in line 217, "..vegetation communities changed towards more forest environemt"?
Changed.

If I am right, the melting of the glaciers mentioned in lines 220-221 are not well supported by the data – this would be shown by high detritic input into the lake during deglaciation, which is not the case (see 7e, highest forest, highest evaporation and lowest detritic input). Not clear…
At the transition from cold to warm periods, the Ca/K ratio shows high detritic input at Lake Van during cold/dry periods (glacials). At the beginning of terrestrial temperate intervals the melting of the glaciers is clearly visible by negative isotope values (up to -4‰ around 240 ka BP suggest low evaporation, high freshwater supply) along with high detrital input into the basin (low Ca/K ratio, ~10, still low forest density).

**Lines 239-241:** This is not clear – please rephrase.

We removed this section.

**Lines 242-245:** The vegetation shift towards more Pinus does not seem to be due higher continentality as stated here. Check Fig. 4, where the peak in Pinus seems to be reached during the lowest seasonal contrast (low summer insolation and high winter insolation – cooler summers and warmer winters).
Thank you very much for this very important comment. We revised the section as follows: 'The ensuing ecological succession of the first penultimate interglacial 
[revised manuscript text omitted]